



# The importance of plant-water stress for predictions of ground-level ozone in a warm world

Tamara Emmerichs[1,3] , Yen-Sen Lu[2,3], and Domenico Taraborrelli[1,3]

[1]Institute of Energy and Climate Research, IEK-8: Troposphere, Forschungszentrum Jülich, Jülich, Germany
[2]Jülich Supercomputing Centre, Forschungszentrum Jülich GmbH, 52425 Jülich, Germany
[3]Center for Advanced Simulation and Analytics (CASA), Forschungszentrum Jülich, Jülich, Germany

**Correspondence:** Tamara Emmerichs (t.emmerichs@fz-juelich.de)

**Abstract.** Evapotranspiration is important for Earth's water and energy cycles as it strongly affects air temperature, cloud cover and precipitation. Leaf stomata are the conduit of transpiration and thus their opening is sensitive to weather and climate conditions. This feedback can exacerbate heat waves and droughts and can play a role in their spatio-temporal propagation. Therefore, the plant response to available water is a key element mediating vegetation-atmosphere interactions. Sustained high

temperatures strongly favor high ozone levels with significant negative effects on air quality and thus human health. Our study assesses the process representation of evapotranspiration in the atmospheric chemistry model ECHAM/MESSy. Diverse water stress parametrizations are implemented in a stomatal model based on $CO_2$ assimilation. The stress factors depend on either soil moisture or leaf water potential and act directly on photosynthetic activity, mesophyll and stomatal conductance. Overall, the new functionalities reduce the initial overestimation of evapotranspiration in the model globally by more than one order

of magnitude which is most important in the Southern Hemisphere. The intensity of simulated warm spells over continents is significantly enhanced. With respect to ozone, we find that a realistic model representation of plant-water stress depresses uptake by vegetation and enhances its photochemical production in the troposphere. These effects lead to a general increases in simulated ground-level ozone which is most pronounced in the Southern Hemisphere over the continents. The uncertainties for plant dynamics representation due to too shallow roots can be addressed by more sophisticated land surface models with

multi-layer soil schemes. In regions with low evaporative loss, however, the representation of precipitation remains the largest uncertainty.

## 1 Introduction

The response of plants to water availability is crucial for climate models since it determines the plant activity and thus photo-

synthesis and transpiration over vegetated land surface. Besides evaporation from open water and soil surfaces, transpiration by plants is with 60-75 % the main contributor to evaporation and transpiration (*ET*: water returned from land to the atmosphere) (Seneviratne et al., 2010). Its strength depends on vegetation coverage, surface wetness, and the availability of soil





water for vegetation root uptake for transpiration. Evapotranspiration (often also termed as terrestrial evaporation, *ET*) in turn has multiple impacts on the hydrological, energy and biogeochemical cycles (Sellers et al., 1997; Seneviratne et al., 2010; Vicente-Serrano et al., 2022; Wang and Dickinson, 2012). A decrease of *ET* in response to land drying reduces the flux of latent heat to the atmosphere, which leads to increased air temperature and decreases the likelihood of rainfall (e.g., Seneviratne et al., 2010).

A scarcity of soil water (water lower than a critical threshold), strengthens the physical plant-water stress limiting the transpiration mediated by the stomata (plants' pores). The resulting change in latent heat flux (of vaporization, $\lambda$) decreases the likelihood of rainfall (Miralles et al., 2019). These conditions, which are predicted to increase due to climate change, could potentially amplify droughts and heatwaves (Kala et al., 2016). Thus, the water availability of plants is a key to realistically represent such weather extremes in the Earth system models (e.g. review by Miralles et al. (2019)). In particular, heatwaves are projected to increase under climate change and thus land-atmosphere coupling gains in importance (Domeisen et al., 2022). Furthermore, terrestrial energy fluxes have become even more sensitive to vegetation over the last decades as Forzieri et al. (2020) found in an observational data set from 1980 to 2016.

Most models use an empirical reduction factor dependent on the volumetric soil moisture content to represent the response of plants to dryness currently (see review by Rogers et al. (2017)). However, this factor does not reproduce the plant response to dryness realistically . Instead, parametrizations based on the independent leaf water potential ($\psi$) perform better (Verhoef and Egea, 2014). Leaf water potential is a vital variable to describe the plant dependence on water, the chemical potential gradient from the root zone to the leaves (Klein, 2014; Sellers et al., 1997) and e.g. Paço et al. (2013) define it as one of the most reliable plant-water stress indicators. The inclusion of $\psi$ in stomatal models is consistent with the hypothesis that stomata regulate transpiration rates in order to avoid cavitation in the xylem. The water potential strongly modulates stomatal conductance at the evaporating sites within the leaf. This is a well established theoretical assumption for modelling transpiration (Tuzet et al., 2003, and references therein).

Yet, studies do not determine whether the plant-water stress acts on photosynthesis or directly modifies the stomatal conductance, which depends on the opening of the stomata (see reviews by De Kauwe et al. (2013); Rogers et al. (2017)). Thus, models differ largely in this regard. Keenan et al. (2010) have shown that neglecting the water stress acting only on photosynthesis significantly overestimate the stomatal opening. Applying the stress factor only to the stomatal conductance could not explain the observed reduction of the assimilation rate in the plant, which is often much larger than the decrease in the stomatal conductance. Further, measurement studies (Drake et al., 2018; Zhou et al., 2013; Egea et al., 2011; Keenan et al., 2010) agree that the water stress acts on the stomata as well as on non-stomatal processes in plants. Thus, the sole application of the water stress to the photosynthesis as done in e.g. the Community Land Model (CLM, Kennedy et al. (2019)) is not sufficient. Egea et al. (2011) has found that drought stress also has a detrimental effect on the mesophyll conductance, which regulates the diffusion between the sub-stomatal internal cavities to the chloroplasts.

We use the global atmospheric chemistry model ECHAM/MESSy Atmospheric Chemistry (EMAC) (Jöckel et al., 2016) to investigate the multiple feedbacks involved and assess the uncertainty related to the evapotranspiration representation from land. This model is widely used to address the simulation and prediction of atmospheric chemistry and tackle global air quality



issues. As part of the Chemistry–Climate Model Initiative (CCMI) (Jöckel et al., 2016), the model community also contributes to climate research. Here, we explore multiple plant-water stress formulations regarding uncertainties and variability, firstly

implemented in EMAC. We assess the performance of the different sensitivity studies at global scale against plant transpiration and evaporation data provided by the GLEAM model and the EUMETSAT satellite, respectively. The consequences of changing the plant-water stress factor for ground-level air pollution are investigated in the next section. We also assess the impact of a changed plant-water response on evapotranspiration in a condition with $2xCO_2$ to account for the global warming. This paper closes with a general discussion of the approach and the model and a comprehensive summary of the results.

## 2  Methods

### 2.1  Model description

#### 2.1.1  Atmospheric model

We use the ECHAM/MESSy atmospheric chemistry model where MESSy (v2.55; Jöckel et al., 2010) provides a flexible infrastructure for coupling processes to build comprehensive Earth system models (ESMs). This is utilised here with the

fifth-generation European Centre Hamburg general circulation model (ECHAM5,version 5.3.02; Roeckner et al., 2003) as the atmospheric general circulation model. To reproduce the large-scale model dynamics, (i.e jet stream) the horizontal winds (divergence, vorticity) are nudged towards reanalysis data of ERA5 by Newtonian relaxation. The model thermodynamics, on the other hand, can freely respond to the process modifications implemented in this study (see Sect. 2.1.3). We perform (dynamical) simulations with 3-hourly instantaneous and average output for each plant-water stress parametrization at meso-

scale (T106: 1.12 ° or ≈ 60km, middle atmosphere) in the period 2017/2018. The warm spell metric is calculated from a dynamical simulation at T42 (2.79 ° or ≈ 300km) covering 1979-2008. To assess the impact on air pollution (see Sect. 3.5) we conduct two chemistry simulations (T106, 2017/2018). Two additional chemistry simulations comprise the $CO_2$-doubling experiments.

#### 2.1.2  Soil and Land representation

The soil water dynamics are represented by a first-generation bucket model including one layer for the water storage (Delworth and Manabe, 1988; Seneviratne et al., 2010). The soil wetness results from the amount of precipitation, snowmelt, evapotranspiration, runoff, and drainage calculated by ECHAM5. The interception of precipitation is calculated for one canopy ('big leaf') layer. Surface runoff is from the overflowing soil water reservoir (Delworth and Manabe, 1988; Roeckner et al., 2003). The initial state is prescribed by geographically varying field capacity which significantly determines the model performance

(Hagemann, 2002; Robock et al., 1998). The data used here were compiled from the most recent global distribution of major ecosystem types made available by the U.S. Geological Survey (Hagemann, 2002). The vegetation density (leaf area index, LAI in [$m^2$ $m^{-2}$]), used to scale the leaf stomatal conductance to the canopy level, is prescribed with a 10-daily time-series observed by the Ocean and Land Colour Instrument (OLCI, visible imaging push-broom radiometer) onboard the Sentinel-3



platform at the Copernicus Land service at an original grid of 1 km (Thépaut et al., 2018). This represents a realistic prod-

uct according to the reported LAI range of 0-6 (Xiao et al., 2017). This data set replaces the climatology used in EMAC as standard.

### 2.1.3 Evapotranspiration and terrestrial photosynthesis

The process of evapotranspiration partially depends on the opening behaviour of the stomata (Katul et al., 2012). Thus, the calculation of evapotranspiration incorporates the stomatal conductance ($g_s$). As already described by Schulz et al. (2001), in

ECHAM the model formulation is based on the Monin-Obukov stability theory:

$$ET = -L_v \rho C_h |\mathbf{v}| \beta(q_a - h q_{sat}(T_s, p_s)) \qquad \beta = [1 + C_h |\mathbf{v}| \cdot 1/g_s]^{-1} \tag{1}$$

where $L_v$ is the latent heat of vaporisation, $\rho$ the density of air, $|\mathbf{v}|$ the absolute value of the horizontal wind speed and the $C_h$ the transfer coefficient of heat. The later two variables translate to $r_a = 1/(C_h |\mathbf{v}|)$. $q_{sat}$ and $q_a$ are the saturation-specific and the atmospheric specific humidity, respectively. The relative humidity $h$ at the surface limits the evapotranspiration from bare

soil. $\beta$ determines the ratio of transpiration between water-stressed plants ($\beta < 1$) and well-watered plants ($\beta = 1$) (Giorgetta et al., 2013; Schulz et al., 2001). The weighted sum of the evapotranspiration over land, water and ice yields the final value. Transpiration is accounted by only a part of equation 1, namely where *ET* is weighted by taking the vegetation fraction in each grid box. The stomatal conductance is calculated by a photosynthesis scheme ($A_{net}$-$g_s$), which is based on Calvet (2000) and is implemented in the IFS model (ECMWF, 2021). This approach describes the photosynthesis process and its dependence on

$CO_2$, temperature and soil moisture (Jacobs, 1994) treating the plants as mixed crops. Currently, ECHAM/MESSy does not distinguish between different land cover types. The photosynthesis model is based on net assimilation rate of $CO_2$ ($A_n$) in the plant varying with environmental conditions ($Env$) and the $CO_2$ concentration outside the leaves ($C_s$, [kg $CO_2$ m$^{-3}$]) and inside the cavities ($C_i$, [kg $CO_2$ m$^{-3}$]) to yield the stomatal conductance ($g_s$):

$$g_s = \frac{A_n(Env)}{C_s - C_i(Env)} \tag{2}$$

The radiation- and $CO_2$-limited scheme are considered for the calculation of net assimilation rate ($A_n$). The saturation of photosynthetic capacity $A_m$ at high light intensities is calculated as follows:

$$A_m = A_{m,max}[1 - \exp(-g_m(C_i - \Gamma)/A_{m,max})] \tag{3}$$

with $A_{m,max}$ being the maximum photosynthetic capacity, $g_m$ the mesophyll conductance, the compensation point at 25 °C $\Gamma = 42$ [ppm] (for mixed crops). The two schemes are combined afterwards to yield a smooth function for $A_n$, which is further

described in ECMWF (2021). $g_m$ is a function of temperature and the mesophyll conductance at 25 °C where the latter involves two different factors for the water state of the atmosphere and the plant-water stress factor (for low and high vegetation) based on a non-linear, empirical expression by Calvet et al. (2004).




### 2.1.4 Water Stress Functions

We investigated several water stress functions and implemented them in the stomatal conductance scheme. The dependence is
commonly parameterised by a fraction of the actual soil water status limited to the availability and the plant wilting (Rogers
et al., 2017). Based on the bucket model used in EMAC, the default function (*REF*) and the multiple application (described
later, *DEFmulti*) employs the actual soil wetness ($W_s$, [m]), the critical available water ($W_{crit}$, [m]) and the wetness at the
wilting point of plants ($W_{pwp}$, [m]) that the plant cannot extract water below this level according to Schulz et al. (2001):

$$f(W_s) = \begin{cases} 1 & W_s(t) \geq W_{crit}(= 75\% F_c) \\ \frac{W_s(t) - W_{pwp}}{W_{crit} - W_{pwp}} & W_{pwp} < W_s(t) < W_{crit} \\ 0 & W_s(t) \leq W_{pwp}(= 35\% F_c) \end{cases} \tag{4}$$

The wilting point depends on soil and vegetation properties such as the soil texture and plant functional type, which is
however only considered indirectly by initialising field capacity ($F_c$) data and therefore introduces a certain amount of uncer-
tainty. This motivates the usage of the original plant-water stress formulation (*noWP*) by Delworth and Manabe (1988), which
considers the critical soil wetness as the solely restriction for plants:

$$f(W_s) = \begin{cases} 1 & W_s(t) \geq W_{crit}(= 75\% F_c) \\ \frac{W_s(t)}{W_{crit}} & W_s(t) < W_{crit} \end{cases} \tag{5}$$

For both parametrizations, the water stress function $f(W_s)$ is considered in the calculation of the mesophyll conductance and
the maximum atmospheric water deficit (in a non-linear way) (Calvet et al., 1998, 2004). Instead of using a soil moisture
dependent function further, we apply the plant-water stress on the $\psi$ according to the findings of Verhoef and Egea (2014). This
is calculated according to Millar et al. (1971), similarly to the formulation employed in Zhang et al. (2003):

$$\psi = -0.395 - 0.043 \cdot Temp_a \tag{6}$$

where $Temp_a$ is the air temperature (in [°C]). The stress factor (*LWPfrac*) is calculated (similarly to Eq. 4) according to Zhang
et al. (2003):

$$f(\psi) = \begin{cases} 1 & \psi \geq \psi_{io} \\ \frac{\psi - \psi_{crit}}{\psi_{io} - \psi_{crit}} & \psi_{io} > \psi > \psi_{crit} \\ 0 & \psi \leq \psi_{crit} \end{cases} \tag{7}$$

where $\psi_{io} = -0.74$ MPa is the leaf water potential at initial reduction, and $\psi_{crit} = -2.75$ MPa the leaf water potential at final
stomatal closure (Verhoef and Egea, 2014).



However, by evaluating the several stomatal models, Sabot et al. (2022) shows that an exponential dependency of $\psi$ is more suitable (*LWPexp*):

$$f(\psi) = \begin{cases} 1 & \psi \geq 0 \\ e^{s_{Med} \cdot \psi} \end{cases} \tag{8}$$

where $s_{Med} = 2\ MPa^{-1}$ is a sensitivity parameter. We further implemented the more sophisticated stress factor used in the common Community Land Model (CLM5, (Kennedy et al., 2019)) as reference (*CLM5*):

$$f(\psi) = \begin{cases} 1 & \psi \geq 0 \\ 2^{-(\frac{\psi}{p50})^{c_k}} \end{cases} \tag{9}$$

where the water potential at 50 % loss of stomatal conductance ($p50 = -1.75$, in [MPa]) and a vulnerability parameter ($c_k = 2.95$) are used. Please note that in CLM5 the soil matric potential is used. However, the leaf water potential can be used as a proxy (Kozlowski et al., 1991; Verhoef and Egea, 2014).

A quantitative limitation analysis by Egea et al. (2011) found that for a realistic model representation water stress should act at least on the biochemical capacity and stomatal conductance and alternatively also on the mesophyll conductance. In most ecosystem models, however, only biochemical or stomatal limitations are included. Therefore, we apply the plant-water stress in case *DEFmulti*, *LWPfrac*, *LWPexp* and *CLM5* linearly to the stomatal and the mesophyll conductance as well as to the photosynthetic activity of plants.

An overview of all parametrizations used as plant-water stress factor in the calculation of stomatal conductance is given in Table 1.

## 2.2 Observational data

### 2.2.1 EUMETSAT

The observational data for evapotranspiration was generated by the European Organisation for the Exploitation of Meteorological Satellite (EUMETSAT) with the second generation of geostationary Meteosat satellites which cover the domain of Europe, Africa and most of South America at 3 km spatial resolution. The Spinning Enhanced Visible and Infrared Imager (SEVIRI) radiometer operating (among others) on board obtains the radiation components at the surface. This data together with further biophysical parameters and soil moisture data from remote sensing, recent land-cover information from the ECOCLIMAP land cover database and meteorological fields from numerical weather prediction drive a physical model of energy exchange between the soil-vegetation-atmosphere systems. By this, the flux [in mm h$^{-1}$] of water evaporated at the Earth-atmosphere interface (soil, vegetation, water bodies) and transpired by vegetation through stomata (as a consequence of photosynthetic





| Case | Plant-water stress factor | | current study (original study) |
|---|---|---|---|
| *noWP* | $f(W_s) = \begin{cases} 1 & W_s(t) \geq W_{crit}(=75\%F_c) \\ \frac{W_s(t)}{W_{crit}} & W_s(t) < W_{crit} \end{cases}$ | (1) | applied in $g_m$ calculation (to final $g_s$) |
| *REF* | $f(W_s) = \begin{cases} 1 & W_s(t) \geq W_{crit}(=75\%F_c) \\ \frac{W_s - W_{pwp}}{W_{crit} - W_{pwp}} & W_{pwp} < W_s < W_{crit} \\ 0 & W_s(t) \leq W_{pwp}(=35\%F_c) \end{cases}$ | (2) | applied in $g_m$ calculation (to final $g_s$) |
| *DEFmulti* | as *REF* (1,3) | | multiplicative factor to $g_m$, $g_s$, $A_{max}$ |
| *LWPfrac* | $f(\psi) = \begin{cases} 1 & \psi \geq \psi_{io} \\ \frac{\psi - \psi_{crit}}{\psi_{io} - \psi_{crit}} & \psi_{io} > \psi > \psi_{crit} \\ 0 & \psi \leq \psi_{crit} \end{cases}$ | (4) | multiplicative factor to $g_m$, $g_s$, $A_{max}$ (to $g_s$) |
| *LWPexp* | $f(\psi) = \begin{cases} 1 & \psi \geq 0 \\ e^{s_{Med} \cdot \psi} \end{cases}$ | (5) | multiplicative factor to $g_m$, $g_s$, $A_{max}$ (to the slope of the sensitivity of $g_s$ to $A_n$) |
| *CLM5* | $f(\psi) = \begin{cases} 1 & \psi \geq 0 \\ 2^{(-\frac{\psi}{p50})^{c_k}} \end{cases}$ | (6) | multiplicative factor to $g_m$, $g_s$, $A_{max}$ |

**Table 1.** Parametrizations for plant-water stress used here, originally by Schulz et al. (2001) (1), Delworth and Manabe (1988) (2), Verhoef and Egea (2014) (3), Zhang et al. (2003) (4), Sabot et al. (2022) (5), CLM5,Kennedy et al. (2019) (6) with $g_m$, $g_s$, $A_{max}$ being the mesophyll conductance,and stomatal conductance, the maximum photosynthetic capacity. $W_s$, $W_{crit}$,$W_{pwp}$ are the actual soil wetness, critical soil wetness and soil wetness at wilting point, respectively. $F_c$ is the field capacity (maximum holding capacity of soil moisture). $\psi$, $\psi_{crit}$ and $\psi_{io}$ are the actual leaf water potential, the critical value, the value at final stomatal closure, respectively. $c_k$, $p50$ and $s_{med}$ are a vulnerability parameter, water loss at 50 % stomatal closure and sensitivity parameter, respectively.

processes) is calculated within a Soil-Vegetation-Atmosphere Transport model (SVAT) (saf, 2018):

$$ET = 3600 \frac{LH_T}{L_v}, \qquad LH_T = \frac{L_v \rho}{(r_a + r_s)}[q_{sat}(Temp_s) - q_a(Temp_a)] \qquad (10)$$

where $LH_T$ is the latent heat flux of transpiration in $[W/m^2]$, $L_v$ the latent heat of water vapor in [J kg$^{-1}$], $\rho$ the air density[kg m$^3$], $r_a$ and $r_s$ are the aerodynamic and stomatal resistances (inverse of the conductances), $q$ the specific humidity

and $q_{sat}(T_s) - q_a(T_a)$ atmospheric saturation deficit in [kg/kg]. This product have been downloaded from the website of the EUMETSAT land surface analysis (LSA SAF) consortium (https://landsaf.ipma.pt/ChangeSystemProdLong.do?system= LandSAF+MSG&algo=DMET, last access: 29.06.2023) at a time interval of 3 hours (original frequency: 30 min). For comparison with the model results, the downloaded dataset was regridded to the spatial grid of EMAC. The product validation report found a general accuracy of 20-25 %, equivalent to the accuracy of measurements. Main uncertainties may stem from

the physical formalism of the algorithm, the errors of the input data, surface heterogeneity and sensor performance among others (saf, 2018).



### 2.2.2 GLEAM

The Global Land surface Evaporation: the Amsterdam Methodology (GLEAM) model estimates the evaporative flux over land by assimilating satellite observations. The land evapotranspiration is the sum of the bare soil, short vegetation, and tall vegetation in each grid box. The soil water content of multiple layers (depending of the land type) is calculated by a water balance between the input snowmelt and rainfall (minus interception). Thereby, surface soil moisture observations from satellites are assimilated (with the Kalman filter approach) at daily time step based on its uncertainty. The Priestly-Taylor equation calculates the potential latent heat flux $\lambda E_p$ $[MJm^{-2}]$:

$$\lambda E_p = \alpha \frac{\Delta}{\Delta + \gamma}(R_n - G) \tag{11}$$

as a function of the net radiation ($R_n$, daily observational data) and the ground-heat flux ($G$). $\Delta$ is the slope of the temperature/saturated vapor pressure curve (in $[k\ Pa\ K^{-1}]$). The division by the latent heat of vaporisation $\lambda$ yields the potential evaporation ($E_p$ in [mm]). For optimal environmental conditions, $\alpha = 0.8$ and $\alpha = 1.26$ at tall and short vegetation (or bare soil) are used, respectively. An evaporative stress ($S$) is used to convert $E_p$ to actual transpiration ($T$ in $[mmday^{-1}]$, over vegetation):

$$T = SE_p \tag{12}$$

S is parameterised separately for tall and short canopies as well as for bare soil (then eq.12 yields bare soil evaporation) based on the observed soil moisture conditions and vegetation optical depth. The canopy interception loss ($I$) is estimated in a separate module based on observations of daily rainfall, snow depth, tall canopy fraction and lightning climatology and parameters for canopy cover, canopy storage, mean rainfall and evaporation rate during saturated canopy conditions. To account for conditions with wet canopy where water is evaporated (and not intercepted) the factor $\beta = 0.07$ is introduced. An extra module estimates the snow and ice sublimation for the snow-covered pixels (no stress) where $\alpha = 0.95$. The evaporation from lakes and rivers is not included. Further details can be found in Miralles et al. (2011). The data was downloaded from the ftp server after registration https://www.gleam.eu/#downloads, last access: 24.07.2023).

### 2.2.3 TROPOSIF

Solar-induced chlorophyll fluorescence ($SIF$), an electromagnetic signal emitted by the chlorophyll of assimilating plants and not used for photosynthesis, can be observed with remote sensing. This can be a proxy for photosynthetic activity because the $SIF$ signal responds to perturbations by environmental stress (Maes et al., 2020). However, the estimation requires high spectral resolution and advanced retrieval schemes since the emissions contribute only a small fraction to the radiance. The TROPOMI (TROPOspheric Monitoring Instrument) instrument aboard the Copernicus Sentinel-5 Precursor mission, launched in October 2017, measures Top-of-the-Atmosphere radiances. By inversion of a linear forward model these are fitted in the far-red spectral region. $SIF$ estimates from the 743-758 nm window are the most robust against atmospheric effects like cloud contamination. The L2B product used here (SIF dataset from TROPOMI: TROPOSIF) combines all observations at the





single orbits within one ungridded netCDF4 file (NOVELTI et al., 2021). The evaluation with other $SIF$ products showed a
general consistency in terms of level and amplitude of the retrieved SIF, and seasonality, for vegetated surfaces. The indicative

error threshold for the definition of spatio-temporal bins is 0.2 mW m$^{-2}$ steradian$^{-1}$ nm$^{-1}$ value (about 10 % of the peak
$SIF$ values observed globally) (Guanter et al., 2015). This translate to 0.064 mm day$^{-1}$ of transpiration. In addition, the
data product includes a quality flag which is used here for individual quality assurance. The data can be downloaded at http:
//ftp.sron.nl/open-access-data-2/TROPOMI/tropomi/sif/v2.1/l2b/ (NOVELTI et al., 2021; Guanter et al., 2015). According to
Maes et al. (2020) the $SIF$ data can be converted to the latent heat flux of transpiration ($LH_T$ in $[W/m2]$):

$$LH_T = 61.4 \cdot SIF \tag{13}$$

Using the latent heat of water vapor ($L_v = 1.5 \cdot 10^6$ in [J kg$^{-1}$]) gives the transpiration [mm day$^{-1}$]:

$$T = LH_T/L_v \cdot 3600 \tag{14}$$

To compare this dataset to the EMAC model we sample the instantaneous output along the satellite orbit at 13:30 UTC.

| Estimation method | Plant transpiration | Evapotranspiration |
|---|---|---|
| EMAC | considers $\beta$ only for the vegetation fraction | $ET = -L_v\rho C_h|\mathbf{v}|\beta(q_a - hq_s(Temp_s, p_s))$<br>$\beta = [1 + C_h|\mathbf{v}|R_{stom}]^{-1}$ |
| Satellite observations by EUMETSAT | not provided | $ET = 3600\frac{LE}{L_v}$<br>$LE = \frac{L_v\rho}{(r_a+r_s)}[q_{sat}(Temp_s) - q_a(Temp_a)]$ |
| GLEAM model driven by satellite observations | $T = SE_p$ | $ET = T + I - \beta I$ |
| Estimate from solar-induced fluorescence by TROPOMI | $LH_T = 61.4 \cdot SIF$<br>$T = LH_T/L_v \cdot 3600$ | not provided |

Table 2. Formulae for plant transpiration and evapotranspiration from EMAC and the used observational datasets.

# 3  Results and Discussion

## 3.1  Plant-water stress and transpiration

The stress functions summarized in Table 1 yield a variety of different plant-water stress and thus transpiration. Figure 1
provides a first overview of how the response functions vary with proxies of water stress (soil moisture and leaf water potential).
Lowering 'volumetric' soil moisture (soil wetness divided by the field capacity) linearly increases the plant-water stress for
the cases *REF* and *DEFmulti* (black line) until the wilting point (35 % of the field capacity) is reached. With the *noWP*
function (gray line), contrarily, plants experience a weaker stress with drying soil, which, however, can increase up to the





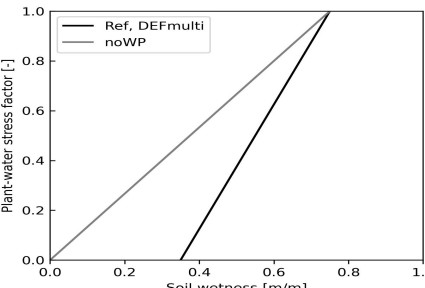
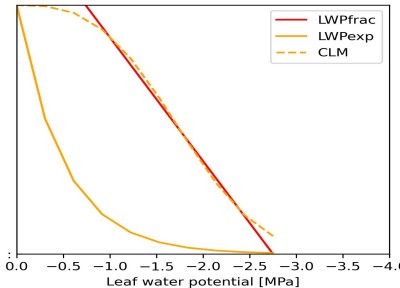

**Figure 1.** Plant-water stress factor vs. (volumetric) soil wetness (left) and leaf water potential (right) of described parametrizations.

point of stomatal closure (stress factor$= 0$). The functions *LWPfrac* and *CLM5* show mostly a linear increase of the stress with increasing water demand (more negative $\psi$). The *CLM5* function covers also the $\psi$ range between 0 and -1 [MPa] where the response is much weaker. *LWPexp* is a simple exponential function with a steep increase of the stress response for $\psi$ from 0 and -1 [MPa]. In comparison, for most plant species Verhoef and Egea (2014) observed a sigmoidal dependency of

plant water stress on soil water (their Figure 1). The recent modelling study by Harper et al. (2021) applied a function with a simple quotient depending on soil moisture similar to the functions *REF* and *DEFmulti*. Model improvements were obtained by replacing the soil moisture with the soil matric potential (Harper et al., 2021), for which $\psi$ applied in *LWPfrac* can be used as a proxy (Kozlowski et al., 1991; Verhoef and Egea, 2014). Early observations of increasing stomatal conductance with a increase of $\psi$ (to lower negative values, see Figure 2B in Sellers et al. (1997)) are in general agreement with these results.

We explore the changes on global and regional scales using spatial (weighted) means for different regions: Europe (oceanic), South America Monsoon (tropical monsoon), Arabian Peninsula (hot arid), African Savanna, boreal forest (continental), East Asia (warm temperate moist). The sensitivity analysis of *noWP* and *DEFmulti* simulations shows only small local changes in transpiration (within the monthly range of variance), impacting the annual estimate only by $\pm$10-15 %. This is because neglecting the wilting point decreases the plant-water stress ($f_{W_s}$) by only 10 % in all dry vegetated regions (dry climate:

$W_s < 0.35 * F_c$, see Seneviratne et al. (2010)) and thus transpiration is only marginally affected.

Figure 2 shows the simulated annual mean maximum photosyntetic capacity ($A_{m,max}$) and transpiration ($T$) and the respective changes. The global distribution (simulated by *REF*) follows the spatial distribution of air temperature and $CO_2$ concentration in the leaf cavities. Until the up-scaling of stomatal conductance to the canopy level (see ECMWF (2021), eq. 8.123) the intermediate calculations, e.g. for $A_{m,max}$, are at leaf level. Thus, the distributions over non-vegetated areas like

the Saharian are masked out here. Transpiration (Figure 2b) additionally depends on atmospheric moisture, which explains its maxima in the tropical rainforests. The multiple application of the default stress factor (to $g_m$, $A_{m,max}$, $g_s$: *DEFmulti*) leads to small decreases of $A_{m,max}$ (Figure 2c) in dry areas (SM$< W_{pwp}$, soil-moisture limited). Thus, transpiration is not significantly changed (Figure 2d, max=0.5).

The impact of the plant-water stress functions based on leaf-water potential (e.g. *LWPfrac*) is more widespread in vegetated

areas since the parametrization is temperature driven. $A_{m,max}$ and also the daily transpiration decreases significantly by 1-



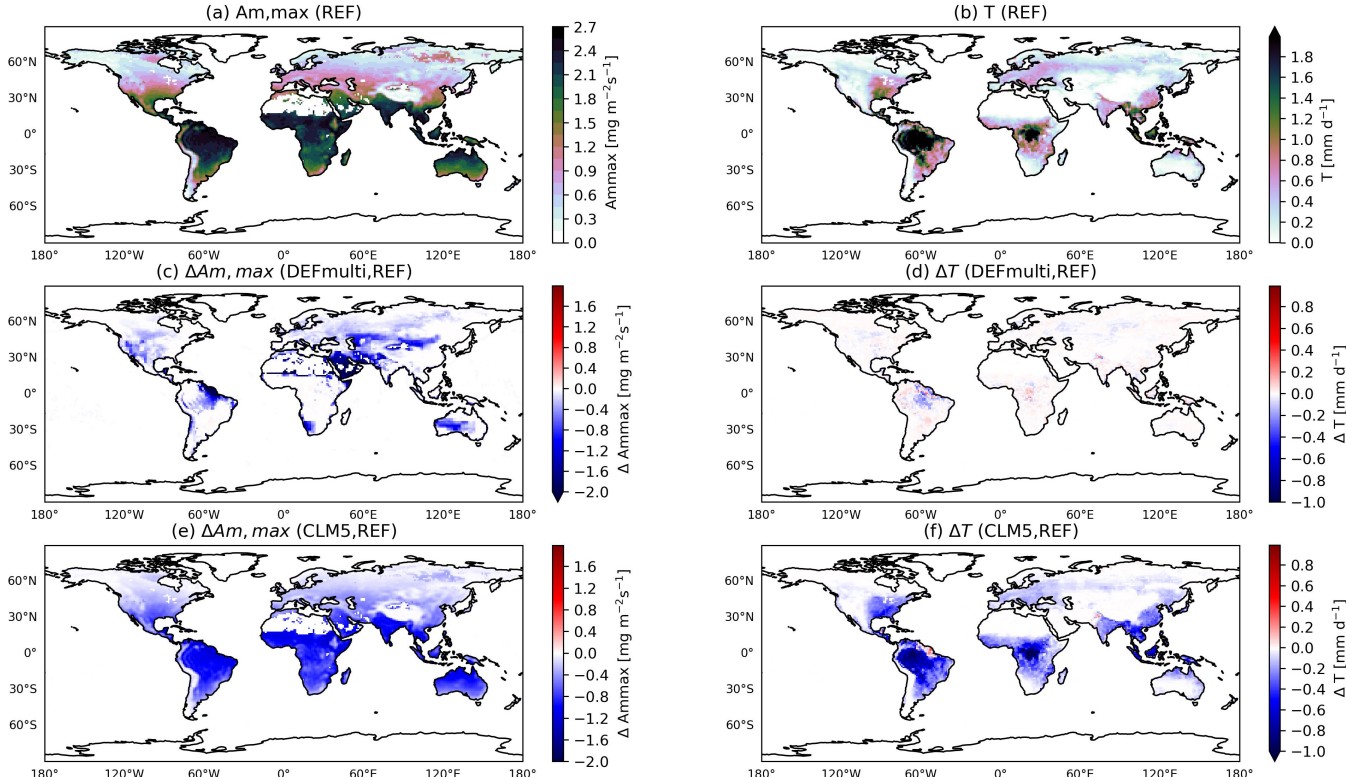

**Figure 2.** Annual mean maximum assimilation rate ($A_{m,max}$) (a), transpiration ($T$) (b) and respective changes to *DEFmulti* (c,d) and *CLM5* (e,f).

2 mm day$^{-1}$ which is highest in the tropical rainforest (Figure 2f). This can be reasoned by the radiation maximum in the inner tropics which leads to a higher influence of the 30 % increase of the plant-water stress and subsequent decrease of the maximum photosynthetic capacity (Figure 2e) and mesophyll conductance (not shown here) in the tropics compared to SH continents. With the start of the boreal summer in May/June the impact spreads out to Europe and the US while it's limited to

the evergreen tropical forests on the SH. Note, that also the final stomatal conductance is lowered again by the stress factor. The changes of the sensitivity simulations *LWPexp* and *CLM5* (not shown here) have the same spatial distribution only a minor different change of the plant-water stress and subsequent variables among each other which means that the linear fraction and the exponential formulation can be interpreted similarly. All three stress functions introduce an additional dependence of the modelled transpiration to air temperature (except in the arid climate). In fact, this slows down the increase of transpiration with

rising temperature. Accordingly, the amplitude of the diurnal cycles decreases (Figure 3 when introducing the multiple stress factor application (*LWPfrac*, *LWPexp*, *CLM5*). On the other hand, the cycle of plant-water stress show firstly variations during day which is an observed phenomena according to Xiao et al. (2021). In contrast to *LWPfrac* and *CLM5* which predict not only the same $\psi$ but also the same $f(\psi)$, *LWPexp* estimates a higher (negative) $\psi$ in most regions (shown in Figure 3). This can



be explained via the temperature-transpiration feedback expected in dry climates (ARP and African savanna). In addition, the
simple exponential function in *LWPexp* yields a stress factor close to zero and thus unrealistically shuts down the mesophyll
conductance and the photosynthetic activity in contrast to *LWPfrac* and *CLM5*.

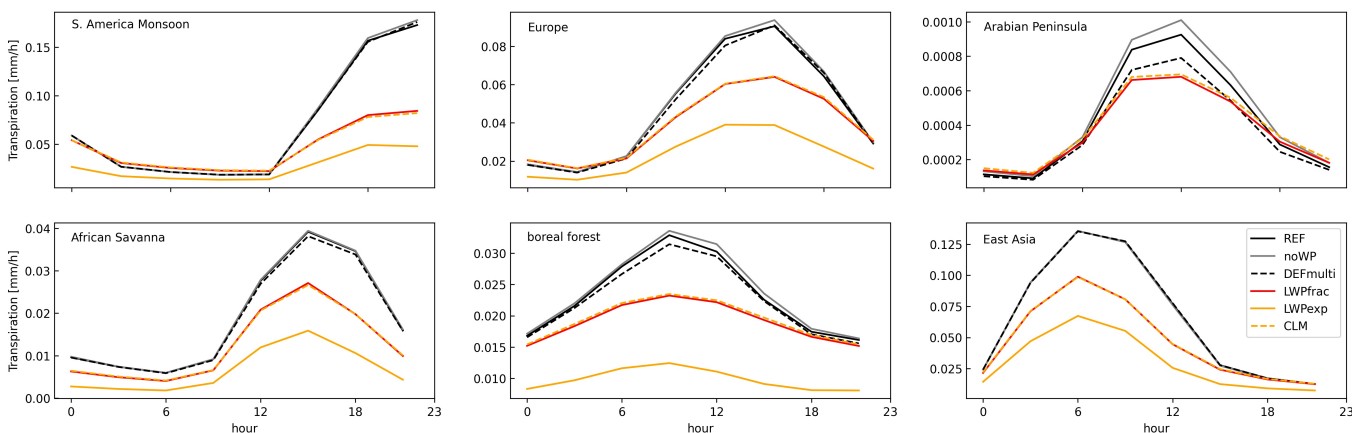

**Figure 3.** Regional mean diurnal cycle of transpiration in boreal summer.

## 3.2 Global estimates of transpiration

All EMAC simulations show a realistic spatial variation of annual transpiration (Figure 2b). However, the low VR values
globally (Table 3) indicate that the simulated variability is lower (VR<1) compared to the GLEAM dataset. This cannot be
attributed to an oversimplification of the modelled process because GLEAM is based on the Priestley-Taylor equation, an
empirical equation dependent on solar radiation and temperature, compared to the physical-based Penman-Monteith approach
used in EMAC (Table 2. The reference simulation of EMAC with the standard plant-water stress overestimates the global
average transpiration calculated with GLEAM by 46 mm yr$^{-1}$ (16 %, Table 3), which is well within the uncertainty range
of the GLEAM product ($\pm$ 136 mm yr$^{-1}$). The *LWPfrac* and *CLM5* stress factors correct for this overestimation regionally.
The global average, however, the new model estimate of 276/277 mm yr$^{-1}$ is lower than the GLEAM estimate. Compared to
the GLEAM uncertainty, all model simulations show a higher 1 $\sigma$ (standard deviation) range indicating a higher uncertainty
which e.g. could be attributed to the representation of precipitation in the model. In GLEAM, instead, precipitation stems
from satellite observations (s. 2.2.2). A lower $1\sigma$ in the sensitivity simulations based on the leaf water potential indicate an
improvement due to neglecting the uncertain soil moisture data usually used in the model. Utilising the transpiration estimate
from the TROPOSIF data yields a good comparison with the (monthly mean) model predictions (only low underestimation)
over areas with high transpiration (e.g. Europe, East Asia) in spring and late autumn. Under strong drought conditions, solar
induced fluorescence by plants decouples from transpiration (Maes et al., 2020) and thus the linear relationship between *SIF*
and *T* (applied here) is not valid anymore e.g. during boreal summer (Martini et al., 2022). Compared to GLEAM (masked
for the TROPOSIF region) however, the TROPOSIF dataset predicts a lower daily transpiration during spring and higher





transpiration during autumn. The seasonality of SIF strongly follows the growing season on the NH which might induce some mismatches.

| Datasets | Transpiration ($1\sigma$) [ mm yr$^{-1}$] | NAE | VR |
|---|---|---|---|
| GLEAM | 329.1 ($\pm$ 68) | - | - |
| *REF* | 375.7 ($\pm$ 98) | 5.00 | 0.08 |
| *noWP* | 379.6 ($\pm$ 100) | 5.59 | 0.07 |
| *DEFmulti* | 370.1 ($\pm$ 97) | 9.80 | 0.08 |
| *LWPfrac* | 277.2 ($\pm$ 77) | 4.85 | 0.11 |
| *LWPexp* | 166.9 ($\pm$ 45) | 10.57 | 0.22 |
| *CLM* | 276.2 ($\pm$ 76) | 4.89 | 0.11 |

**Table 3.** The global estimates of transpiration ($1\sigma$ - standard deviation), normalised absolute error (NAE) and the variance ratio (VR: $\frac{var(mod)}{var(obs)}$, accounting for grid boxes with more than 1 % vegetation.

The multi-model *ET* estimate of 18 CMIP6 models (1980-2014, general increase of ET) and the observation-based T/ET ratio of 64 % by Pan et al. (2020) yield a global transpiration of 384 mm yr$^{-1}$. From this, it can be concluded that all model estimates in our study predicted annual transpiration reasonably well. The only exception is the sensitivity simulation *LWPexp*

showing an unrealistic strong reduction thus a high normalised absolute bias (NAE) which is likely due to the choice of parameters constraining the stress factor significantly (s. 8). For the further impact assessment in this study, we use the stress factor *LWPfrac* since it overall shows the best performance (slightly better than the *CLM5* factor).

### 3.3 Contribution to global evapotranspiration

The contribution of transpiration to the total *ET* varies in time and space with vegetation and soil characteristics (Wang and
Dickinson, 2012; Cao et al., 2022; Lian et al., 2018). This spatial variability is reflected by GLEAM and EMAC whereas especially the estimates in Europe and Africa mismatch (Figure 4). The dominance of soil evaporation over transpiration in dry (non-vegetated) regions as reported by Lian et al. (2018) is here also shown in the African desert by a low T/ET ratio (in GLEAM and EMAC) and non-vegetated parts of China (EMAC). Also, the low T/ET ratio in northernmost areas (partly snow-covered) of Canada and Siberia (see Lian et al. (2018)) is only captured by EMAC. In humid regions, especially the
tropics, evapotranspiration is driven by transpiration. The contribution can reach up to 87 % over densely vegetated regions. For comparison, observations in the Amazonian tropical forest indicate an average T/ET ratio of 0.7 (Wang and Dickinson, 2012; Zhang et al., 2017). This can be consistently represented by EMAC (Figure 4b) although the sensitivity simulations, e.g. *LWPfrac* and *CLM5*, partly reduce the T/ET ratio too much in the south of the South America continent (Figure 4c,d). According to the simulated and observation-based estimates of T/ET by Lian et al. (2018) (their Figure 1a), all EMAC sim-
ulations represent too low values in most parts of U.S. suggesting a dry model bias. For the central U.S., Dong et al. (2022)





indeed confirms that unbiased estimates of summertime daily maximum temperature could be achieved only with a T/ET ratio of 0.7. Contrarily, GLEAM shows higher values of the T/ET ratio for the east coast of the U.S. as well as for the SH continents, Europe, and Asia. The incorrect E-T partitioning was identified as an error source of ET estimation in CMIP5 models (Lian et al., 2018).

**Figure 4.** Annual mean ratio of transpiration evapotranspiration by (a) GLEAM, (b) *REF*, (c) *DEFmulti*, and (d) *LWPfrac*).

To assess the model estimation of evapotranspiration we compare with *ET* estimates by GLEAM and EUMETSAT whereas GLEAM shows generally higher estimates (Figure 5a, c). *ET* has its maximum in the tropics while in the high northern latitudes and sparse-vegetated areas (e.g. South African desert) low values occur. The GLEAM estimate of (EUMETSAT-region) *ET* (512 mm yr$^{-1}$) differs by 30 mm yr$^{-1}$ (6 %) from the EUMETSAT value (481 mm yr$^{-1}$) which could be considered to be within the uncertainty range. However, regionally the difference can be large, as much as 50 %. This is most evident in the tropics and consistent with recent studies reporting a large spread and a high uncertainty in model estimates for *ET* at low latitudes due to the parametrization of the root water uptake (Pan et al., 2020). According to literature values by (e.g., Elnashar




et al., 2021), who calculated an annual *ET* of 540 mm yr$^{-1}$ (for 2018), the GLEAM estimate is the most consistent with literature values. Thereby, the models usually differ by 200 mm yr$^{-1}$ which is about twice the spread of estimates by single models (minima and maxima) (Wang et al., 2021).



**Figure 5.** Annual evapotranspiration (*ET*) of (a) GLEAM, and its difference to (b) the *CLM5* sensitivity simulation (*CLM5*-GLEAM), (c) EUMETSAT and (d) the difference to the the *CLM5* sensitivity simulation.

The global average of annual *ET* predicted by EMAC with the different plant-water stress parametrizations is about 425-480 mm yr$^{-1}$. ET predicted by the *CLM5* sensitivity simulation, which reproduces transpiration the best (see Sec. 3.2, together with *LWPfrac*), compares well with the GLEAM annual values. Mainly in some coastal areas like East U.S., NE Amazon considerable differences occur which could be reasoned by neglected sub-scale hydrology at the coasts (Figure 5b). Compared to EUMETSAT, EMAC (as well as GLEAM) estimates a higher annual mean *ET* in tropical rain forests whereas in tropical monsoon climate region too low values are simulated compared to EUMETSAT (Figure 5d). This pattern of differences suggests precipitation as a reason since these two climate types differ essentially by the amount of precipitation. This is consistent with the known precipitation bias of the ECHAM5 climate model (see Figure 7 in Stevens et al. (2013)). Both, EMAC and



EUMETSAT underestimates the GLEAM global *ET* where, however, more than 50 % of the mismatch occurs outside the EUMETSAT region. The difference cannot always be considered to be within the model variability of 20 % due to the model

net radiation depending on the choice of forcing data (Badgley et al., 2015). One reason for the underestimation is likely the neglect of diffuse radiation impact in big-leaf models, as used here, enhancing photosynthesis and evapotranspiration (Wang et al., 2022; Knohl and Baldocchi, 2008). Furthermore, representing also deep plant roots would ensure a more realistic water holding capacity and avoid a drying out of the soil in the tropical rainforests (Hagemann and Stacke, 2015).

## 3.4 Impact on air temperature

The changes in *ET* have significant impacts on air temperature. Here, we compare the temperature predicted by *REF* to the one by *LWPfrac*. As expected, from a decrease of *ET*, i.e. less cooling, high daily maximum air temperature values increase, shown in Figure 6 for warm spells in 2018. We define warm spell conditions as a period of at least 3 consecutive days when daily mean temperature exceeds the 95 % percentile of the daily mean temperature of the reference period (1979-2008) (Nairn and Fawcett, 2014). In fact, the difference of the actual temperature to to the climatological percentile (termed 'excess heat factor'

in Nairn and Fawcett (2014) which is a measure of intensity of warm spell conditions increases by 1.5K in Europe and 4K in South Africa, in the East U.S. and the Amazon forest due to the changed plant-water stress function of *LWPfrac*. The global mean air temperature in the lowest model layer ($\approx$ 60m) increases by 2K. Our results are consistent with recent studies, (e.g., Kala et al., 2016), highlighting the role of stomatal stress in the amplification of heatwaves especially affecting the intensity of warm spells and heat waves (Barriopedro et al., 2023).

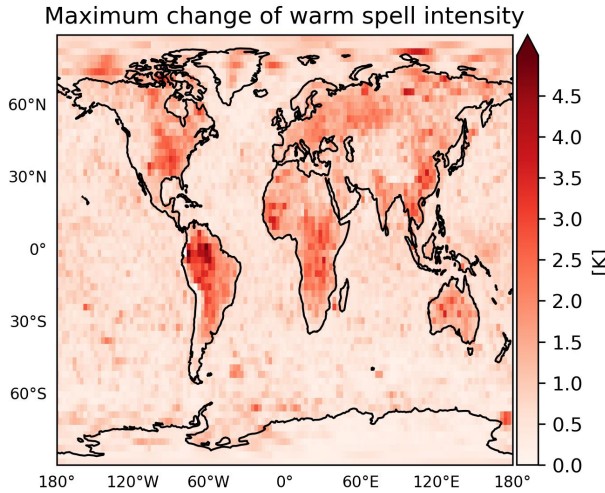

**Figure 6.** The maximum annual difference of warm spell intensity in 2018 due to the plant water stress function.





## 3.5 Impacts on air pollution

The different representations of plant-water stress affect air pollution mainly by influencing 1) dry deposition fluxes of ozone and 2) meteorological controls on photo-chemistry. Figure 7 shows the respective changes for ozone ($O_3$) which is a major air pollutant threatening human health as well as the productivity of plants. Figure 7a shows that the dry deposition of $O_3$ in *LWPfrac* is decreased by up to 25 %, compared to *REF*, in the tropics and subtropics where dry deposition exerts a strong control on air composition due to high vegetation density. Similar changes apply to precursors with similar characteristics as $O_3$ which then contributes to the increase of $O_3$ mixing ratio (Emmerichs et al., 2021). Furthermore, the reduced *ET* in most vegetated regions exacerbates the atmospheric moisture deficit by which the stomata are additionally stressed. The annual mean chemical production and loss terms (Figure 7b,c) are only enhanced in the SW of South America (by up to 10 %) although the increased plant-water stress leads to a significant temperature increase in the entire tropical regions (see previous section) which is known to favour $O_3$ production (Pusede et al., 2015). The increase of $O_3$ production, shown, here follows the increase of OH and $HO_2$ ($HO_x$) production but it is limited to western Amazon. That is because, in the inner tropical rainforest (Amazon, Congo) the isoprene mixing ratio, an important $O_3$ precursor, decreases (Figure S1b) due to increased loss by hydroxyl radical (OH) although isoprene emissions are enhanced by higher temperatures (Guenther et al., 2006). The change of the $O_3$ loss has the same magnitude but is more widespread than the change of the $O_3$ production driven by a relative acceleration of $NO_x$ and $HO_x$ chemistry. These effects then lead to an increase of the net $O_3$ loss in the Amazon basin which is overcompensated by the decreased $O_3$ uptake by vegetation. Thus, annual mean surface $O_3$ is increased in the tropics and subtropics by up to 10 % (Figure 7d). This enhances the tropospheric $O_3$ burden by 5 Tg per year.

## 3.6 Future scenario

A simulation with the double $CO_2$ concentration (*futureLWPfrac*) was performed to investigate the role of the new plant-water stress factor in future climate conditions. Besides perturbing the energy balance at the top of the atmosphere, $CO_2$ affects the plant sensitivity to water stress in our simulations. Increasing $CO_2$ has a two-fold impact on the plants behaviour. While it leads to an increased photosynthetic activity, the stomatal conductance is reduced by an average of 40 % ($g_s$, Figure 8a). Vicente-Serrano et al. (2022) reports a decrease of 22 % (on average) in stomatal conductance from multiple experiments by doubling only $CO_2$. We also can confirm these findings for equatorial and tropical forests in our simulation. The transpiration of plants decreases in response to increasing $CO_2$ in these regions due to the dominant decrease of $g_s$ as reported by Vicente-Serrano et al. (2022). In our simulations, however, the impact of the future conditions on $g_s$ is more widespread since the changed climatic conditions reduce the relative humidity almost world-wide and thus stress the plants. The decrease of $g_s$ by 30 % linked to the new plant-water stress function is strengthened by the enhanced $CO_2$. However, this dominates the *ET* only on a daily basis while the annual sum increases by 30-100 mm yr$^{-1}$ in response to an increased evaporative demand. As a consequence, 2m temperature is almost doubled (Figure 8b) and the relative humidity drops (not shown). These changes are linked to the 20-50 % increase of solar irradiation (correlation) due to less low-level clouds. Pollard and Thompson (1995) also reports on conducting a doubling $CO_2$ scenario leading to an increase in stomatal conductance, temperature and specific







**Figure 7.** The relative change between *LWPfrac* and *REF* of the annual mean of (a) $O_3$ dry deposition, (b) chemical $O_3$ production, (c) chemical loss and (d) surface $O_3$ mixing ratio.

humidity which reduces relative humidity and cloudiness. Nevertheless, to assess the overall climatic impact of the multiple interactions between terrestrial vegetation and $CO_2$ also the changing vegetation would have to be considered. However, such

an assessment is far more complex and highly uncertain (Vicente-Serrano et al., 2022).

## 4 General discussion

### 4.1 Default model parametrization

In models, *ET* is estimated either by the physically-based Penman-Monteith (PM) approach (state-of-the-art) or the empirical Priestley-Taylor (PT) equation. The latter one (used in GLEAM) assumes that *ET* only depends on solar radiation and temper-

390 ature neglecting wind speed, relative humidity and vapour pressure deficit. But because of the link to air temperature, estimates





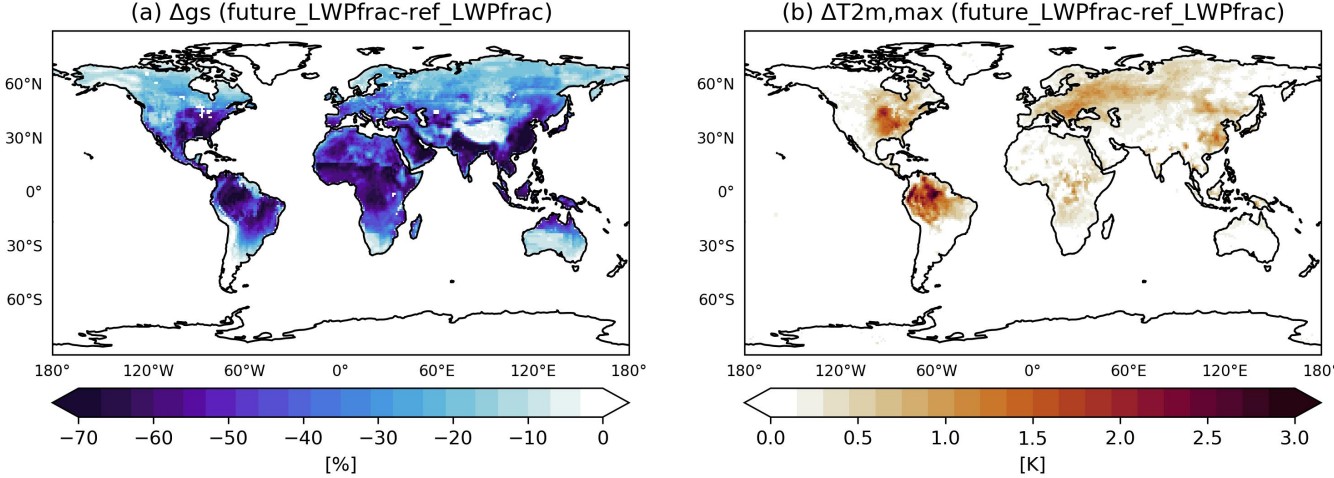

**Figure 8.** (Boreal) Summer mean change of stomatal conductance (a) and daily 2m maximum temperature (b) when comparing *LWPfrac* in normal and future conditions.

by the PT approach show a high correlation to values estimated by the PM equation expect in dry conditions and in areas with relatively high wind speed (Utset et al., 2004). The key variable for the common parametrization of the water stress in plants is the soil moisture described in EMAC by the simplistic but conventional bucket model. A bucket model has been used e.g. in the JSBACH land surface model for a long-time (Boone et al., 2004). The inclusion of the surface resistance term in EMAC as
the so-called "second-generation models" yields a better comparison of estimated evapotranspiration rates with observations than utilizing 'pure' bucket models (Sellers et al., 1997). However, the lack of soil water holding capacity in the bucket model leads to an immediate remove of water and thus to an unrealistically low soil water in areas with deep roots e.g. tropical forests (Hagemann and Stacke, 2015), despite the thickness of subsurface layers. Nevertheless, the multi-model evaluation by Robock et al. (1998) found no significant improvements of sophisticated soil models with multiple layers and even vegetation dynamics
like the CLM or NOAH-LSM over the bucket scheme. More recently, Dong et al. (2022) concluded that most CMIP6 models simulate a warm bias in mid-latitude summer because of incorrect partitioning *ET* in canopy transpiration and soil evaporation due to a shallow soil. Moreover, even small differences of the input field capacity data can have large effects on the simulated *ET* (Hagemann and Stacke, 2015).

### 4.2 More sophisticated models, remaining uncertainties and future recommendations

Boone et al. (2004) shows that sophisticated land surface models (LSMs) agree with each other regarding latent heat flux and total runoff. Nevertheless, we note that comparing different LSMs is very difficult because of the different model components, parameterizations, and choice of associated parameters. Also, many LSMs only represent shallow soil with a depth down to maximum 2m (Pan et al., 2020) and therefore cannot account for the storage capacity of the soil in the tropical forests as shown by Hagemann and Stacke (2015). For the second-generation LSMs Pitman (2003), which calculate transpiration and



soil moisture across multiple layers, the predicted soil moisture is somewhat better than with the bucket model. However, when compared to observations, LSMs show a large spread in performance (Shao and Henderson-Sellers, 1996). This is certainly due to, but not limited to, the use of different schemes for simulating surface fluxes and soil moisture. Generally, the needed spin-up time by LSMs with deep soil schemes is often not affordable, especially for climate simulations. Using in addition a groundwater model ((e.g., Kollet and Maxwell, 2008)) can improve the simulation of the water budget and the groundwater-land surface interactions (Rahman et al., 2014) but strongly increase the required computational resources.

The most recent model intercomparison CMIP6 shows on average an overestimated *ET* by the models compared to an observational dataset. However, the CMIP6 ensemble mean underestimates *ET* in regions of high evapotranspiration, such as the Amazon basin, central Africa, and southeast Asia but overestimates *ET* in regions with low evapotranspiration, such as the Sahara desert, the Middle East, southwest Australia, and the Andes Mountains (Wang et al., 2021). A multi-model comparison by of *ET* estimates Pan et al. (2020) shows that the uncertainty is largest in the Amazon basin, where the standard deviation of LSM estimates is more than 2 times larger than that of benchmark estimates. The potential source of uncertainty is the root water uptake. Also, the model representation of LAI dynamics or water movement in the soil might cause this uncertainty (Pan et al., 2020). In arid and semiarid areas, precipitation is a key uncertainty factor for estimates of evapotranspiration (Pan et al., 2020).

## 5 Conclusions

We have investigated the significance of plant-water stress for the predictions of ground-level ozone concentrations in a warm(er) world. This study has focused on the improvement and assessment of the evapotranspiration simulated by the atmospheric chemistry model EMAC. We confirm that evapotranspiration is a key process driving the moisture cycling in the atmosphere affecting the global distribution of temperature and warm spell intensity. We also find that plant-water stress has a significant impact on the photo-chemistry and uptake of trace gases by vegetation. For that, we have applied multiple plant-water stress factors, which strongly reduce stomatal activity, and have assessed the impacts at local and global scales. Specifically, we find that:

- The EMAC model represents the spatial variability of transpiration reasonably well

- The global estimates of transpiration are within the literature range whereas a simple exponential dependence on leaf water dependence (*LWPexp*) induces a too strong reduction

- The use of stress factors based on leaf water potential lowers the amplitude of the transpiration diurnal cycle but strengthens the model sensitivity to temperature

- The *E/T* partitioning is generally well simulated by EMAC but in regions like the East U.S. the T/ET ratio is too low, probably due to the dry model bias

Close to pollution sources, tropospheric ozone is projected to increase in the future as consequence of the climate warming. This is often referred to as the 'ozone-climate penalty' (Rasmussen et al., 2013). However, a recent multi-model projection



suggests a climate benefit on a global average (Zanis et al., 2022). As many uncertainties remain, a recent analysis call for a re-examination of the link between extreme events and ground-level ozone (Fu and Tian, 2019). Our results highlight the importance of evapotranspiration and plant-water stress for the predictions of air pollution during heat waves and droughts.

These extreme events are projected to be more frequent and intense (Domeisen et al., 2022). The magnitude of the effects assessed in this study are model-specific. Nevertheless, they provide a general guidance for assessment and improvement of atmospheric chemistry models without a state-of-the-art description of land surface processes.

*Code and data availability.* The Modular Earth Submodel System (MESSy) is continuously further developed and applied by a consortium of institutions. The usage of MESSy and access to the source code is licensed to all affiliates of institutions which are members of the

MESSy Consortium. Institutions can become a member of the MESSy Consortium by signing the MESSy Memorandum of Understanding. More information can be found on the MESSy Consortium Website http://www.messy-interface.org. The code used in this study is included in the current devel branch of the MESSy repository. The simulation results are archived at the Jülich Supercomputing Centre (JSC) and are available on request. The EUMETSAT *ET* data is available from the website of the EUMETSAT land surface analysis (LSA SAF) consortium (https://landsaf.ipma.pt/ChangeSystemProdLong.do?system=LandSAF+MSG&algo=DMET. The GLEAM data can be provided

by a registered user via a ftp server (https://www.gleam.eu/#downloads, last access: 24.07.2023). The TROPOSIF data can be downloaded at http://ftp.sron.nl/open-access-data-2/TROPOMI/tropomi/sif/v2.1/l2b/ (NOVELTI et al., 2021; Guanter et al., 2015).

*Author contributions.* All authors designed and frequently discussed the concept of the study, TE implemented the code changes supported by DT and Y-SL. The simulations, the data processing and the data analysis were done by TE. All authors wrote and reviewed the manuscript.

*Competing interests.* The authors declare that they have no conflict of interest.

*Acknowledgements.* The authors gratefully acknowledge the Gauss Centre for Supercomputing e.V. (www.gauss-centre.eu) for funding this project by providing computing time on the GCS Supercomputer JUWELS (Jülich Supercomputing Centre, 2019) and by the John von Neumann Institute for Computing (NIC) and provided on the supercomputer JURECA (Jülich Supercomputing Centre, 2021) at Jülich Supercomputing Centre (JSC). The EUMETSAT product was provided by the EUMETSAT Satellite Application Facility on Land Surface Analysis (Trigo et al., 2011).The TROPOSIF products were generated by the TROPOSIF team conducted by NOVELTIS under the European

Space Agency (ESA) Sentinel-5p+ Innovation activity Contract No 4000127461/19/I-NS (NOVELTI et al., 2021; Guanter et al., 2015). The regridding script was adapted from the work of UWE Raschers group at Forschungszentrum Jülich. This work was supported by funding from the Federal Ministry of Education and Research (BMBF) and the Helmholtz Research Field Earth & Environment for the Innovation Pool Project SCENIC.



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
