# Peer review of "The influence of plant-water stress on vegetation-atmosphere exchanges: implications for ozone modelling"

_EGUsphere, 2023_

## Author Response (AR1)

**1 Answer to Reviewer 1**

*In this article, the authors explore representations of evapotranspiration in the chemistry-climate model EMAC, and how this impacts air temperature as well as air pollution, focusing mainly on tropospheric O3. The remarks below need to be addressed before the manuscript can be accepted for publication.*

We are very grateful to Reviewer 1 for the thoughtful and constructive comments helping us improve the quality of the manuscript. We try to consider all the comments. Please find our reply (in blue) to your comments (black, italic) below.

*This work presented is within the scope of the journal, and I think the results are robust and quite interesting. However, I have two major concerns:*

*1. I think the manuscript has a lot of build-up but then the implications on air quality are not discussed thoroughly. In other words, Section 3.5 needs to be expanded. I listed some suggestions below.*

We agree with Reviewer 1. The section was extended, see comments below.

*2. I had to re-read some text multiple times to understand what the authors wanted to say. I strongly recommend the re-writing of some parts. It would probably be a good idea to send the manuscript for proofreading.*

We thank Reviewer 1 for pointing to this aspect. We will improve the general readability and clarity of the whole manuscript.

**1.1 General comments:**

*While reading the manuscript I came across words written in US English and British English. Please be consistent.*

We now stick to US English.

*I think the manuscript would benefit from having an "Experimental design" section where you describe the experiment setup in detail. i.e., not just the relevant parameterisations, but also the submodules responsible for the land surface, vegetation, etc.*

We add a 'Experimental design' section as section 2.1.4, which describes the set-up in more detail and use the respective lines from Section 2.1.1 (now 2.1):
'We perform dynamical simulations with 3-hourly instantaneous and average output for each plant-water stress parametrization at meso-scale (T106: 1.12 ° or ≈ 60km, middle atmosphere) for the period 2017/2018. The dynamical simulations apply a set of submodules (AEROPT, CLOUD, CLOUDOPT, CONVECT, GWAVE, MSBM, OROGW, ORBIT, QBO, RAD, SUR-FACE, TROPOP, VERTEX), similar to the set up used in Joeckel et al. (2016). The land–atmosphere exchange and vertical diffusion in EMAC is here described by the submodel VERTEX (Emmerichs et al., 2021). The key functionalities of VER-TEX are explained in Section 2.1.2. The warm spell metric is calculated from a dynamical simulation at T42 (2.79 ° or ≈ 300km) covering 1979-2008. To assess the impact on air pollution (see Section 3.7) we conduct two chemistry simulations (T106, 2017/2018). These simulations employ additionally submodules describing emissions of atmospheric species (OF-FEMIS, ONEMIS, BIOBURN, LNOX), gas exchange submodels (DDEP, AIRSEA). Chemical kinetics is calculated in the gas-phase by the submodule MECCA (Sander et al., 2019) and in cloud droplets by the submodule SCAV (Tost et al., 2006), JVAL (Sander et al., 2014). The chemical mechanism includes the basic gas-phase chemistry of ozone, methane, and odd nitrogen with in total 310 reactions and 155 species as in Jöckel et al. (2016). Dry deposition of trace gases to vegetation is calculated according to the multiple resistance scheme which utilises the stomatal resistance calculated in VERTEX. The scheme is used here with six generalised land types. The vegetation canopy is represented as one system; i.e. the detailed struc-ture and plant characteristics are neglected (one big-leaf approach). The leaves are horizontally oriented and the leaf density is uniformly vertically distributed (Kerkweg et al., 2006; Emmerichs et al., 2021). Further information regarding the submodules

can be found in Jöckel et al. (2010, 2016). Two additional chemistry simulations comprise the $CO_2$-doubling experiments. To reproduce the large-scale model dynamics, e.g. jet stream, the horizontal winds (divergence, vorticity) are nudged towards reanalysis data of ERA5 by Newtonian relaxation with an e-folding time of 6-24 hours for all wavenumber truncations in the spectral space similarly to van Garderen et al. (2021). Temperature and pressure are not nudged. This selective nudging is applied for conducting storyline simulations (Shepherd et al., 2018). This allows the model thermodynamics to respond freely to the process modifications implemented in this study.'

*It is not clear what vegetation representation is being used. It was only later in L357/358 that I realised that you have no vegetation in your model. This should be made more clear. The suggestion above would help to clarify this.*

At the end of Section 2.1.2, we added the sentence 'EMAC contains no dynamic vegetation model. Vegetation is represented as a "big leaf" affecting exchanges of water, energy and trace gases depending on leaf area index, soil moisture and other parameters. This is described in detail in Section 2.1.3.'

*Figure captions are way too short. Generally, one should be able to understand the figure completely from the caption without referring to the main text (i.e. self-describing). E.g., Fig. 2 - "Annual mean maximum assimilation" Assimilation of what?*

There, $CO_2$ assimilation is meant. We extend this caption with this information.

*Fig. 3 was not discussed enough in the text. Please update the figure's caption (see above), and also provide the geographical extent of the areas considered.*

We discuss Fig.3 verbosely in line 253-267 including the differences and features of the regional plots. The text is rewritten with shorter sentences and more clear structure to the following:
'Only a minor different change of the plant-water stress and subsequent variables among each other is seen in the regional plots (Fig. 3). Thus, the the linear fraction and the exponential formulation can be interpreted similarly. All three stress functions based on leaf water potential (*LWPfrac*, *LWPexp*, *CLM5*) introduce an additional dependence of the modelled transpiration on air temperature (except in the arid climate). On the one hand, this dampens the increase of transpiration with rising temperature. Accordingly, the amplitude of the diurnal cycles decreases (Figure 3). On the other hand, the diurnal cycle of plant-water stress show firstly variations with temperature during day which is an observed phenomena according to Xiao et al. (2021). In contrast to *LWPfrac* and *CLM5* which predict not only the same $\psi$ but also the same $f(\psi)$, *LWPexp* estimates a higher (negative) $\psi$ in most regions (shown in Figure 3). This can be explained via the temperature-transpiration feedback expected in dry climates (ARP and African savanna). In contrast to *LWPfrac* and *CLM5*, the simple exponential function in *LWPexp* yields a stress factor close to zero unrealistically shutting down the mesophyll conductance and the photosynthetic activity. The analysis of *noWP* and *DEFmulti* simulations shows only small local changes in transpiration (within the monthly range of variance) which impacts the annual estimate only by $\pm$10-15 %. This is because neglecting the wilting point decreases the plant-water stress ($f_{W_s}$) by only 10 % in all dry vegetated regions (dry climate: $W_s < 0.35 * F_c$, see Seneviratne et al. (2010)) and thus transpiration is only marginally affected.'
The caption of Figure 3 is extended, and regional definitions are added.

*I find some of your colour bars confusing. When using a diverging color bar, please decide whether you want the min/max to be pointy or not, but be consistent! The min/max of the colour bars is misused in most of the figures presented.*

We agree with Reviewer 1. All diverging color bars were adapted

*Fig. 4 - Fix the colour bar for subplot (a). Caption - Which panels are from EMAC and which are not?*

Panel b-d are from EMAC, this information was added in the caption

[Figure]

**Figure 1.** Annual mean relative change (*LWPfrac-REF*) of OH (a) and isoprene (b) mixing ratio at the surface (for regions with isoprene>50 ppt).

*Fig. 8 - Color bars. Why do you have a change in stomatal conductance over deserted regions, e.g., the Sahara desert? Please explain this, and if the values are negligible consider applying a mask.*

Until the up-scaling of stomatal conductance to the canopy level (see ECMWF (2021), eq. 8.123) the intermediate calcula-
100  tions, are at leaf level. Thus, we apply also here a mask now to neglect non-vegetated areas like the Sahara desert.

*Section 3.5: In the supplement, you show a strong decrease in isoprene mixing ratios. Could you also show the absolute difference not just the relative (%) change?*

105  Yes, we added the absolute difference for isoprene mixing ratio.

*From Fig. 6 I would expect a strong increase in isoprene emissions given the increase in surface temperatures. Would be nice to dig in deeper here and quantify the increase in surface isoprene emissions and see how this increase compares to the*
110  *decrease in isoprene concentrations from enhanced OH oxidation. Also, provide details on the isoprene emissions and sensi-*

[Figure]

**Figure 2.** (a) Regional mean diurnal cycle of O3 in the Amazon (Monsoon region, definition in Fig. 3) and (b) linear regression of the absolute difference (*LWPfrac-REF*] formaldehyde (HCHO) with O3 surface levels at the ATTO (Amazon Tall Tower Observatory) site in November 2018 (dry season).

*tivities. Are you using MEGAN?*

We add further details to the text:
'The plant emission activity, as modelled by the MEGAN model (Model of Emissions of Gases and Aerosols from Nature)
115 increase with higher temperature up to a value of approximately 40° C (Guenther et al., 2006). The increasing emissions lead to a linear increase of O3. As shown in Figure 2a and b for the Amazon, the O3 increase by 0.34 ppb per 1 ppb increase in formaldehyde (HCHO). HCHO is a direct product of isoprene oxidation with a lifetime of a few hours and thus often used as a proxy for isoprene emissions (Palmer et al., 2003). In the outer tropics, O3 additionally increase with rising soil emissions of nitrogen oxides (NO), which is an important O3 precursor source in remote regions (far from anthropogenic emissions).'
120

*Fig. S1: In some places over Africa (e.g. 20-30 deg S), you show a decrease in the OH radicals but this does not correspond to an increase in isoprene. Why is that? This seems to violate your hypothesis that isoprene concentrations go lower because of increased loss by OH. Similarly, could you explain the hotspot (increase in isoprene) over Antarctica where there are no apparent corresponding changes in OH?*
125

Thank you for mentioning this interesting features. These African places experience significant NO soil emissions. The higher NOx emissions limit the oxidation of plant emissions (compared to other remote regions) (Monks et al., 2015). Isoprene on Antarctica stems from transported emissions over the ocean but this can be treated as insignificant. Thus, we here show only changes where the surface isoprene level exceeds 50 ppt.
130

*Please include some limitations in your study. E.g. No vegetation representation, and the fact that the biosphere and BVOC emissions do not respond to changes in tropospheric chemistry e.g. ozone concentrations.*

The model represents vegetation and related coupling processes. We made clear in the model description that it, however,
135 has no dynamical vegetation model and the LAI as only information. The plant damage by O3, which is probably meant here with the second aspect, is out of the scope of this study and will be published soon in a separate study. The limitations are considered here at the end of Sect. 3.5 as follows:
'The here discussed changes do not include the O3 damage to plants, i.e the biosphere-atmosphere exchange. However, from experiments like by Sadiq et al. (2017) we can learn that an implementation of this response amplifies the O3-vegetation
140 feedback. Since the caused O3-increase limits the plant activity reducing transpiration and dry deposition further which in turn enhances the O3 levels. No clear feedback was found for isoprene emissions while the reduced ecosystem production contributes only minor to the overall feedback.'

**1.2 Minor comments:**

*L26: When writing "e.g." for citations, it is generally expected that you mention more than one study. Please correct all other instances in the manuscript.* we added references in these places or re-formulated to be more concrete

*L29: "(plant's pores)" - I don't think this is needed here.* removed

*L36/37: "Currently" should go at the start of the sentence.* changed

*L51: "non-stomatal processes in plants" - like what?* like mesophyll diffusion and biochemical limitations, added to the text

*L61: "GLEAM" - Is this an acronym? Please define acronyms on their first instance.* defined already in the Section 2.2.2

*L61: More details are needed on the EUMETSAT satellite you are referring to.* The details are described in Section 2.2.1

*L90: Include unit for LAI m2/m2.* included

*L98: Latter not "later"* changed

*164: EUMETSAT was already used. Please define in the first instance.* It is defined in the description of EUMETSAT

*L184/85: Not clear what you mean here. Do you mean the sum of the bare soil, short/tall vegetation evapotranspiration per grid box?* yes exactly

*L193: mmday-1 - why italic?* we write now written units in the text consistently in italic font.

*L246/253: Text not clear. Please consider rewriting this part. "Saharian" - do you mean Sahara desert?* yes, is corrected.

*L250: The distribution of transpiration inf Fig. 2b follows patterns of vegetation distribution e.g. LAI. Why would the additional soil moisture in the tropics explain the geographical distribution?* The atmospheric moisture deficit not the soil moisture is meant

*L302: "African desert" - Sahara desert* changed

*L308: "...in the southern part of South America ....."* different wording

*L344: "to to"* corrected

*L352: "... respective changes in tropospheric ozone...."* rephrased to 'shows the impacts of using the *LWPfrac* plant-water stress on troposheric ozone'

*L392/393: Provide more details on the "bucket model" in EMAC or JSBACH.* The bucket model is already described in Sect. 2.1.2

*L394: JSBACH define the model acronym.* defined now

*L405: "...(LSMs) generally agree with..."* The line was adapted

*L414: Double brackets.* changed

*L401/402: Give more details on the 'ozone-climate penalty'. What is the benefit suggested in Zanis et al. 2022?* We added the sentence:

'However, a recent multi-model projection suggests a climate benefit on a global average, i.e. a decrease of ozone as consequence of global warming.'

**2 Answer to Reviewer 2**

*Emmerichs et al. assessed the implementation of various functions related to plant water stress in the ECHAM/MESSy model and examined their subsequent effects on change in evapotranspiration. The authors also investigated the impact of changes in evapotranspiration on ground-level ozone. While this modelling exercise is valuable for pinpointing the appropriate functions implemented in their atmospheric chemistry model, there are a few drawbacks in the current form of this paper, particularly regarding the overall structure and model-data validation. Here I provide some comments for the authors' and editor's consideration.*

We are very grateful to Reviewer 2 for the thoughtful and constructive comments helping us improve the quality of the manuscript. We try to consider all the comments. Please find our reply (in blue) to your comments (black, italic) below.

**2.1 Major comments:**

*The title of this paper is misleading. It led me, as a reader, to expect a primary focus on how plant-water stress influences ground-level ozone. However, two-thirds of the Results are on testing/comparing the impact of changes in plant-water stress functions on plant transpiration, evapotranspiration, and air temperature. Ozone is only a minor aspect of the findings. It might be more appropriate to adjust the title to better reflect the actual results. If the authors aim to quantify the effect of plant water stress on ozone, the results should provide the correlation between the water stress index/ET/Temperature and ozone concentration.*

We understand the reviewers opinion and adjust the title to:
'The influence of plant-water stress on vegetation-atmosphere exchanges: implications for ozone modelling. '

*2. It is not clear to me from the text how the change in evapotranspiration (ET) will affect ozone concentration through atmospheric chemical processes. I was trying to get some basics from the Introduction and Method but was unsuccessful. Although the author does mention chemical processes in Section 3.5, it would be more effective to introduce this information much earlier to provide a general understanding of how changes in ET could impact ozone. While the relationship may be self-evident in atmospheric chemistry, it is still essential to include basic information in the paper to present a comprehensive story.*

We add basics on tropospheric ozone and its dependence on meteorology to the introduction (see answer of 3. item in 'Minor comments'. The description of O3-related chemistry in Section 3.5 was also extended in response to the first reviewer.

*3. A major concern regarding the model's performance is the absence of model-measurement validation. All data presented in Section 2.2 consist of simulations from other models. No ground-based data were used to validate the ET estimates obtained by modifying the plant-water stress functions in their model. Data from EMUETSAT or GLEAM may provide a robust estimation of ET by validating their performance against ground-based measurements. As an independent model, the outputs from ECHAM/MESSy should also be validated against ground-based measurements. It is feasible to get the ET measurements from the FLUXNET network nowadays. A direct comparison in a 1:1 space, plotting ground-based measurements against model estimates, would provide a clear assessment of the model's performance and illustrate how changes in plant water stress functions impact ET and other model outputs.*

We certainly acknowledge the advances of FLUXNET. However, this global model study can not provide a point-to-point comparison. Although we use relatively high spatial resolution for a chemistry global model, the grid boxes are too large with respect to the footprint region of the ground-based measurements. Spatial heterogeneity of land characteristics is very important for ET and soil moisture at the measurement sites and the global model cannot resolve those scales. To account for the broad spatial coverage of the model results a global observation dataset (TROPOSIF and GLEAM) is better suited. Furthermore, we chose a rather recent study period in line with the focus of our funding project SCENIC. Data from this period, however, is not available for the most sites of FLUXNET.

**2.2 Minor comments:**

*1. The first 4 sentences of the abstract were written in a way that jumps from one topic to another without logical connections.*

To improve the readability we change the text:
'Evapotranspiration is important for Earth's water and energy cycles as it strongly affects air temperature, cloud cover and precipitation. Leaf stomata are the conduit of transpiration. Thus, their opening is sensitive to weather and climate conditions. This feedback can exacerbate heat waves and can play a role in their spatio-temporal propagation.'

to

'Evapotranspiration drives the Earth's water and energy cycles and thus strongly affects air temperature, cloud cover and precipitation. Transpiration through leaf stomata contributes a major fraction. The stomatal opening is sensitive to weather and climate conditions. The weather sensitivity can lead to the amplification of heat waves determining their spatio-temporal propagation.'

*2. "Overall, the new functionalities" What do these new functionalities refer to? No mention in the previous text.*

Here, we refer to the new implemented functions/parametrisations. We will change the wording to 'new parametrisations'.

*3. Before Line 55, there should be a section to introduce the research background on ozone using modelling.*

We agree and add a paragraph on tropospheric ozone:

Tropospheric ozone is a major air pollutant harmful to humans and plants. Its spatial and temporal evolution not only depends on emissions but it is also crucially dependent on meteorological variables like temperature. In fact, radical reactions dominating the $O_3$ formation are enhanced at high temperatures. Also, plant emissions of isoprene, a major ozone precursor, strongly respond to increasing temperature rising exponentially until a temperature of $42°C$ is reached (Guenther et al., 2006). Higher temperatures as well as dryness inhibit dry deposition, an important sink of ozone and its precursors. A major part of dry deposition happens at the stomata during the water-/$CO_2$-exchange of plants (transpiration/respiration). As plants close their stomata to limit the water loss (Katul et al., 2009) ozone uptake is strongly reduced.

In line 60, we add:

For assessing the impact of the different plant-water stress on ozone, we use a comprehensive chemistry with 310 reactions and 155 species in the gas-phase. Anthropogenic emissions are prescribed from reanalysis and CCMI data. Natural emissions of ozone precursors (from lightning, soil and plants) are simulated interactively based on respective measurements and parametrisations (Guenther et al., 2006; Tost et al., 2006; Kerkweg et al., 2006).

*4. In line 55, please provide the full name of MESSy, where it is first introduced.*

We add the full name: '(Modular Earth Submodel System)'.

*5. From Lines 105 to 115, it is not clear what are inputs and outputs from the listed equation. It states (L154) that "The two schemes are combined afterwards to yield a smooth function for An" and then (L108)" to yield the stomatal conductance (gs)". Is An used to calculate gs after An is derived from Am?*

We modify line 99 ff. accordingly referring to more details in the supplement section 1.

Further details of the calculation are provided in the manuscript supplement.

We add to the supplement:

According to the established IFS model, $A_n$ is derived from the saturation level $A_m$ (among others) and is used for the calculation of $g_s$ afterwards. The calculation of the net assimilation rate ($A_n$) distinguishes for a $CO_2$ limiting and the radiation limiting regime which changes at level $A_m$ (from radiation to $CO_2$ limiting regime):

$$A_m = A_{m,max} \left[ 1 - \exp(-g_m(C_i - \Gamma)/A_{m,max}) \right] \tag{1}$$

The maximum photosynthetic capacity $A_{m,max}$ is calculated as follows:

$$A_{m,max}(T_s) = \frac{A_{m,max}(25)Q_{10Am,max}^{(T_s-25)/10}}{(1+e^{0.3(T_{1am,max}-T_s)})(1+e^{0.3(T_{2am,max}-T_s)})} \qquad (2)$$

with $T_{1am,max} = 8°C$, $T_{2am,max} = 38°C$ and $A_{m,max} = 2.2 mg(CO_2)m^{-2}s^{-1}$. The mesophyll conductance $g_m$ is calculated:

$$g_m(T_s) = \frac{g_m(25°C)Q_{10gm}^{(T_s-25)/10}}{(1+e^{0.3(T_{1gm}-T_s)})(1+e^{0.3(T_{2gm}-T_s)})} \qquad (3)$$

with $T_{1gm} = 5°C$ and $T_{2gm} = 36°C$. $T_s$ is the leaf surface temperature (here 2m temperature) and the $Q_{10}$ constant is 2. $g_m(25°C)$ depends on soil moisture and is further described in ECMWF (2021). An exponential transition function represents $A_n$ in dependence on radiation and $A_m$.

*6. The temperature dependence of gm is highly variable. What function is used to describe gm and why in your model?*

This aspect is addressed in the extended model description above.

*7. Line 119 how do you implement the water stress function into the stomatal conductance? Please provide the used function here.*

We add to line L.121/122.

[...] which are given in Sec. S1 of the supplement.
To supplement section S1, we add:
According to Calvet et al. (1998, 2004) plants respond in the a complex way through the mesophyll conductance ($g_m$) to soil moisture stress:

$$g_m(25°C) = g_m^N \frac{f(W_s)}{W_{crit}} \qquad (4)$$

$$g_m(25°C) = g_m^N + g_m^0(25°C)\frac{f(W_s)-W_{crit}}{1-W_{crit}} \qquad (5)$$

where $g_m^0(25°C)$ is a species-dependent parameter (here: 25 $mm\ s^{-1}$). $g_m^N$ is the stressed value of $g_m$ and described in ECMWF (2021).

*8. Figure 2 How the Am,max is derived? It seems unrealistic in central Australia and southern Africa where most of the area is desert with low photosynthesis but model predictions of Am,max are high in those regions in panel a.*

We refer in line 250 to the (newly added) formula of $A_{m,max}$ in the supplement. Line 254, we extend the following:

[...] which depends on the model vegetation mask. $A_{m,max}$ is strongly driven by leaf (2m) temperature which is reflected by the distribution plot in Fig. 2a.

*9. Line 344, delete "to"*

In line 344, we only find 'the' needless and deleted it.

*10. Line 372 Please elaborate on the correlation between stomatal conductance and photosynthesis and how the change in*
320 *$CO_2$ concentration will affect the stomatal conductance and photosynthesis differently.*

We a add the following in line 400:

ECHAM/MESSy does not simulate an interactive carbon cycle, namely the photosynthesis i.e. the net assimilation of $CO_2$ is
325 calculated for simulating the stomatal conductance with a first order dependence scaled by the $CO_2$ deficit between plant cavity
and atmosphere. Several studies reported that an increase of atmospheric $CO_2$ reduces the leaf stomatal conductance varying
from 50 % in dense meadows, to 15 % in broad-leaved forests, and to less than 10 % in coniferous forests. This response is
non-linear because the $CO_2$ stimulation of photosynthesis saturates at high atmospheric $CO_2$. (Vicente-Serrano et al. (2022)
and references therein).

330 **3 Answer to Editor:**

*The reviewers were generally positive about this manuscript's suitability and scientific contribution once their concerns are*
*addressed. The authors have responded positively to the reviewers' comments and questions, so I invite them to submit a revised*
*manuscript based on those responses*

335 We thank the editor for the answer and carefully prepare an improved version of our manuscript.

*Further to the reviewers' comments, I request that the authors take into account these additional comments:*
*1. In the authors' proposed changes to their abstract in response to the comment by Reviewer 2, the first proposed sentence*
*is not correct. The evaporation phase change is the point in the hydrological cycle where a large fraction of absorbed solar*
340 *radiation enters the cycle. But it is solar radiation that "drives". Also, the proposed sentences are at least as disconnected as*
*in the submitted version, perhaps more so. For instance, "Transpiration through leaf stomata contributes a major fraction."*
*Major fraction of what?*

"We regret that the proposed changes are not sufficiently precise and change the four first sentences from:
345 Evapotranspiration is important for Earth's water and energy cycles as it strongly affects air temperature, cloud cover and
precipitation. Leaf stomata are the conduit of transpiration. Thus, their opening is sensitive to weather and climate conditions.
This feedback can exacerbate heat waves and can play a role in their spatio-temporal propagation."
to:
"Evapotranspiration plays a key role for the water and energy fluxes over the continents. A major fraction of the fluxes is pro-
350 vided by plant transpiration which depends on weather and climate conditions. The response of plant transpiration to weather
extremes like droughts can exacerbate the severity of heatwaves."

*2. Please ensure that Reviewer 1's comments about figure captions is addressed (figure and caption should stand alone).*
*The proposed new Fig. 2 does not appear to be properly attributed.*
355

All figure captions have been extended.

*3. At line 93, "The process of evapotranspiration" is not correct. ET is not a single process. Clarify – do you mean tran-*
*spiration?*
360

We change: 'The process of evapotranspiration partially depends on the opening behaviour of the stomata (Katul et al., 2012)'
to:
Transpiration depends on the opening behaviour of the stomata (Katul et al., 2012)

365    *4. Text across lines 97-101 is identical to text in the author's earlier published paper (https://gmd.copernicus.org/articles/14/495/2021/).*
   *Please ensure this is paraphrased.*

   We further paraphrase:
   '[...] where $L_v$ is the latent heat of vaporisation, $\rho$ the density of air, $|\mathbf{v}|$ the absolute value of the horizontal wind speed and $C_h$
370    the transfer coefficient of heat. The latter two variables translate to $r_a = 1/(C_h|\mathbf{v}|)$. $q_{sat}$ and $q_a$ are the saturation-specific and
   the atmospheric specific humidity, respectively. The relative humidity at the surface $h$ limits the evapotranspiration from bare
   soil. $\beta$ determines the ratio of transpiration between water-stressed plants ($\beta < 1$) and well-watered plants ($\beta = 1$) (Giorgetta
   et al., 2013; Schulz et al., 2001).'
   to:
375    [...] where $L_v$ gives the latent heat of vaporisation, $\rho$ represents the density of air. $|\mathbf{v}|$ is the absolute value of the horizontal
   wind speed and $C_h$ describes the transfer coefficient of heat which links via the equation: $r_a = 1/(C_h|\mathbf{v}|)$. $q_{sat}$ and $q_a$ give
   the saturation-specific and the atmospheric specific humidity $h$ is the relative humidity at the surface by which the evapotran-
   spiration from bare soil is limited. At $\beta = 1$, only bare soil evaporation occurs while $\beta < 1$ is used for water-stressed plants
   (Giorgetta et al., 2013; Schulz et al., 2001).
380

   *5. Line 200: "To account for conditions with wet canopy where water is evaporated (and not intercepted)" - Wet canopy*
   *evaporation is usually referred to as interception loss, so "and not intercepted" does not make sense.*

   We change line 200 accordingly to:
385

   Applying a fraction of interception loss ($\beta = 0.007$) ensures that evaporation at wet canopy is only considered once in the
   calculation.

   *6. Consistent with Reviewer 1, please have the revised manuscript independently proofread and edited.*
390

   The language in the manuscript was improved with a proof-reading tool.

**References**

Calvet, J.-C., Noilhan, J., Roujean, J.-L., Bessemoulin, P., Cabelguenne, M., Olioso, A., and Wigneron, J.-P.: An interactive vegetation SVAT model tested against data from six contrasting sites, Agricultural and Forest Meteorology, 92, 73–95, https://doi.org/10.1016/S0168-1923(98)00091-4, 1998.

Calvet, J.-C., Rivalland, V., Picon-Cochard, C., and Guehl, J.-M.: Modelling forest transpiration and CO2 fluxes—response to soil moisture stress, Agricultural and Forest Meteorology, 124, 143–156, https://doi.org/10.1016/j.agrformet.2004.01.007, 2004.

ECMWF: IFS Documentation CY47R3, IFS Documentation, ECMWF, 2021.

Emmerichs, T., Kerkweg, A., Ouwersloot, H., Fares, S., Mammarella, I., and Taraborrelli, D.: A revised dry deposition scheme for land–atmosphere exchange of trace gases in ECHAM/MESSy v2.54, Geoscientific Model Development, 14, 495–519, https://doi.org/10.5194/gmd-14-495-2021, publisher: Copernicus GmbH, 2021.

Giorgetta, M. A., Roeckner, E., Mauritsen, T., Bader, J., Crueger, T., Esch, M., Rast, S., Kornblueh, L., Schmidt, H., Kinne, S., Hohenegger, C., Möbis, B., Krismer, T., Wieners, H., and Stevens, B.: The atmospheric general circulation model ECHAM6: Model description, Reports on Earth System Science, p. 177, 2013.

Guenther, A., Karl, T., Harley, P., Wiedinmyer, C., Palmer, P. I., and Geron, C.: Estimates of global terrestrial isoprene emissions using MEGAN (Model of Emissions of Gases and Aerosols from Nature), Atmospheric Chemistry and Physics, 6, 3181–3210, https://doi.org/10.5194/acp-6-3181-2006, 2006.

Jöckel, P., Kerkweg, A., Pozzer, A., Sander, R., Tost, H., Riede, H., Baumgaertner, A., Gromov, S., and Kern, B.: Development cycle 2 of the Modular Earth Submodel System (MESSy2), Geoscientific Model Development, 3, 717–752, https://doi.org/10.5194/gmd-3-717-2010, publisher: Copernicus GmbH, 2010.

Jöckel, P., Tost, H., Pozzer, A., Kunze, M., Kirner, O., Brenninkmeijer, C. A. M., Brinkop, S., Cai, D. S., Dyroff, C., Eckstein, J., Frank, F., Garny, H., Gottschaldt, K.-D., Graf, P., Grewe, V., Kerkweg, A., Kern, B., Matthes, S., Mertens, M., Meul, S., Neumaier, M., Nützel, M., Oberländer-Hayn, S., Ruhnke, R., Runde, T., Sander, R., Scharffe, D., and Zahn, A.: Earth System Chemistry integrated Modelling (ESCiMo) with the Modular Earth Submodel System (MESSy) version 2.51, Geoscientific Model Development, 9, 1153–1200, https://doi.org/10.5194/gmd-9-1153-2016, publisher: Copernicus GmbH, 2016.

Katul, G. G., Palmroth, S., and Oren, R.: Leaf stomatal responses to vapour pressure deficit under current and CO-enriched atmosphere explained by the economics of gas exchange, Plant, Cell & Environment, 32, 968–979, https://doi.org/10.1111/j.1365-3040.2009.01977.x, publisher: Wiley Online Library, 2009.

Kerkweg, A., Buchholz, J., Ganzeveld, L., Pozzer, A., Tost, H., and Jöckel, P.: An implementation of the dry removal processes DRY DEPosition and SEDImentation in the Modular Earth Submodel System (MESSy), Atmospheric Chemistry and Physics, 6, 4617–4632, https://doi.org/10.5194/acp-6-4617-2006, publisher: Copernicus GmbH, 2006.

Monks, P. S., Archibald, A. T., Colette, A., Cooper, O., Coyle, M., Derwent, R., Fowler, D., Granier, C., Law, K. S., Mills, G. E., Stevenson, D. S., Tarasova, O., Thouret, V., von Schneidemesser, E., Sommariva, R., Wild, O., and Williams, M. L.: Tropospheric ozone and its precursors from the urban to the global scale from air quality to short-lived climate forcer, Atmospheric Chemistry and Physics, 15, 8889–8973, https://doi.org/10.5194/acp-15-8889-2015, 2015.

Palmer, P. I., Jacob, D. J., Fiore, A. M., Martin, R. V., Chance, K., and Kurosu, T. P.: Mapping isoprene emissions over North America using formaldehyde column observations from space, Journal of Geophysical Research: Atmospheres, 108, https://doi.org/https://doi.org/10.1029/2002JD002153, 2003.

Sadiq, M., Tai, A. P. K., Lombardozzi, D., and Val Martin, M.: Effects of ozone–vegetation coupling on surface ozone air quality via biogeochemical and meteorological feedbacks, Atmospheric Chemistry and Physics, 17, 3055–3066, https://doi.org/10.5194/acp-17-3055-2017, 2017.

Sander, R., Jöckel, P., Kirner, O., Kunert, A. T., Landgraf, J., and Pozzer, A.: The photolysis module JVAL-14, compatible with the MESSy standard, and the JVal PreProcessor (JVPP), Geoscientific Model Development, 7, 2653–2662, https://doi.org/10.5194/gmd-7-2653-2014, 2014.

Sander, R., Baumgaertner, A., Cabrera-Perez, D., Frank, F., Gromov, S., Grooß, J.-U., Harder, H., Huijnen, V., Jöckel, P., Karydis, V. A., et al.: The community atmospheric chemistry box model CAABA/MECCA-4.0, Geoscientific model development, 12, 1365–1385, https://doi.org/10.5194/gmd-12-1365-2019, 2019.

Schulz, J.-P., Dümenil, L., and Polcher, J.: On the land surface–atmosphere coupling and its impact in a single-column atmospheric model, Journal of Applied Meteorology, 40, 642–663, https://doi.org/10.1175/1520-0450(2001)040<0642:OTLSAC>2.0.CO;2, 2001.

Seneviratne, S. I., Corti, T., Davin, E. L., Hirschi, M., Jaeger, E. B., Lehner, I., Orlowsky, B., and Teuling, A. J.: Investigating soil moisture–climate interactions in a changing climate: A review, Earth-Science Reviews, 99, 125–161, https://doi.org/10.1016/j.earscirev.2010.02.004, 2010.

Shepherd, T. G., Boyd, E., Calel, R. A., Chapman, S. C., Dessai, S., Dima-West, I. M., Fowler, H. J., James, R., Maraun, D., Martius, O., et al.: Storylines: an alternative approach to representing uncertainty in physical aspects of climate change, Climatic change, 151, 555–571, https://doi.org/https://doi.org/10.1007/s10584-018-2317-9, 2018.

445

Tost, H., Jöckel, P., Kerkweg, A., Sander, R., and Lelieveld, J.: Technical note: A new comprehensive SCAVenging submodel for global atmospheric chemistry modelling, Atmospheric Chemistry and Physics, 6, 565–574, https://doi.org/10.5194/acp-6-565-2006, 2006.

van Garderen, L., Feser, F., and Shepherd, T. G.: A methodology for attributing the role of climate change in extreme events: a global spectrally nudged storyline, Natural Hazards and Earth System Sciences, 21, 171–186, https://doi.org/10.5194/nhess-21-171-2021, publisher: Copernicus GmbH, 2021.

450

Vicente-Serrano, S. M., Miralles, D. G., McDowell, N., Brodribb, T., Domínguez-Castro, F., Leung, R., and Koppa, A.: The uncertain role of rising atmospheric CO2 on global plant transpiration, Earth-Science Reviews, 230, 104 055, https://doi.org/10.1016/j.earscirev.2022.104055, 2022.

Xiao, J., Fisher, J. B., Hashimoto, H., Ichii, K., and Parazoo, N. C.: Emerging satellite observations for diurnal cycling of ecosystem processes, Nature Plants, 7, 877–887, https://doi.org/10.1038/s41477-021-00952-8, number: 7 Publisher: Nature Publishing Group, 2021.

455

---

## Editor Decision (ED1)

**Notes on changes indicated in the response but not made in the manuscript**

The responses to reviewers were not always reflected by the actual changes (or lack thereof) made in the revised manuscript. It seems that the additional text, as presented in the response document, was further edited in the manuscript, making it difficult to reconcile in some instances. Please ensure that these documents are consistent.

Inconsistencies that need attention (line numbers are from the response document):

- Lines 57-59. The text 'EMAC contains no dynamic vegetation model…" does not appear in the revised manuscript.
- Lines 114-117. These three sentences do not appear in the revised manuscript. This is especially important to address because it appears to be the only place where Figure 7 is cited.

**Other technical corrections that should be made**

Line 2. "Thus, their opening…" does not follow very well the previous sentence. Suggest "Leaf stomata are the conduit of transpiration, and their opening…"

Lines 14. Delete "but"?

Line 19 surfaces

Line 21. "with 60-75%" makes no sense here. Do you mean transpiration makes up 60-75% of ET?

Line 23. ET is  already defined. Do you mean to define TE? Only do this if you continue to use the term throughout the paper.

Lines 24-25 and 28-29. There is too much duplication between these.

Line 99. What about non-water-stressed plants?

Line 158. VERTEX is not mentioned in Section 2.1.2.

Line 183. What is the reference (saf, 2018)?

Line 199. Priestly should be spelled "Priestley".

Line 251. "photosyntetic" – mis-spelt.

Line 310. "This is here also shown here…". Please correct.

Figure 7. This is not referenced in the text, although it was proposed to be in the reviewer response document.

Line 386 "… has a two effects …" - delete "a"

Line 390. "… plant transpiration of plants …" – please resolve duplication of "plant/s"

Lines 394-5. I'm sure that 2 m temperature does not double, especially in K units! Please revise.

---

## Author Response (AR2)

**1 Notes on changes indicated in the response but not made in the manuscript**

*The responses to reviewers were not always reflected by the actual changes (or lack thereof) made in the revised manuscript. It seems that the additional text, as presented in the response document, was further edited in the manuscript, making it difficult to reconcile in some instances. Please ensure that these documents are consistent.*
*Inconsistencies that need attention (line numbers are from the response document):*
*Lines 57-59. The text 'EMAC contains no dynamic vegetation model...'' does not appear in the revised manuscript.*
*Lines 114-117. These three sentences do not appear in the revised manuscript. This is especially important to address because it appears to be the only place where Figure 7 is cited.*

We implemented the changes consistently and extent it to "EMAC contains no dynamic land surface model". The text can be found in line 89.

**2 Other technical corrections that should be made**

*Line 2. "Thus, their opening..." does not follow very well the previous sentence. Suggest "Leaf stomata are the conduit of transpiration, and their opening..."*

We changed the text as proposed.

*Lines 14. Delete "but"?*

We deleted this.

*Line 19 surfaces*

The 's' was added to 'surface'.

*Line 21. "with 60-75%" makes no sense here. Do you mean transpiration makes up 60-75% of ET?*

Yes, we changed the sentence accordingly.

*Line 23. ET is already defined. Do you mean to define TE? Only do this if you continue to use the term throughout the paper.*
No, also ET is meant here. We delete the explanation in the brackets.

*Lines 24-25 and 28-29. There is too much duplication between these.*

We agree and removed the sentence in line 29: 'The resulting change in latent heat flux (of evaporation, $\lambda$) reduces the likelihood of rainfall (Miralles et al., 2019).' The explanation 'latent heat flux (of evaporation, $\lambda$) was integrated in the sentence in line 25.

*Line 99. What about non-water-stressed plants?*

The $\beta$ factor is related to the soil moisture and thus is used to constrain the transpiration and stomata opening due to the content of moisture. The application of soil moisture stress can be seen in CLM4.5 (Oleson et al., 2013) and Noah LSM (Niu et al., 2011). For the non-water-stressed plants it can not transpire when there is no water for root-uptake.

*Line 158. VERTEX is not mentioned in Section 2.1.2.*

We added this to line 94: '[....] the model formulation in EMAC (submodel VERTEX) [...]'.

*Line 183. What is the reference (saf, 2018)?*

It is the Product User Manual For Evapotranspiration and Surface Fluxes. We corrected the reference.

*Line 199. Priestly should be spelled "Priestley".*

Corrected

*Line 251. "photosyntetic" – mis-spelt.*

corrected

*Line 310. "This is here also shown here...". Please correct.*

Corrected

*Figure 7. This is not referenced in the text, although it was proposed to be in the reviewer response document.*

Yes, we apologize. The proposed changes are now implemented consistently in the revised manuscript in e.g. line 369.

*Line 386 "... has a two effects ..." - delete "a"*

done

*Line 390. "... plant transpiration of plants ..." – please resolve duplication of "plant/s"*

done

*Lines 394-5. I'm sure that 2 m temperature does not double, especially in K units! Please revise*

Yes that needs correction. We changed the sentence in to:
As a consequence, the 2m temperature increases by up to 3 K (Figure 9b) [...].

**References**

Niu, G.-Y., Yang, Z.-L., Mitchell, K. E., Chen, F., Ek, M. B., Barlage, M., Kumar, A., Manning, K., Niyogi, D., Rosero, E., Tewari, M., and
85    Xia, Y.: The community Noah land surface model with multiparameterization options (Noah-MP): 1. Model description and evaluation
with local-scale measurements, J. Geophys. Res., 116, D12 109, https://doi.org/10.1029/2010JD015139, 2011.

Oleson, K., Lawrence, D., Bonan, G., Drewniak, B., Huang, M., Charles, D., et al.: CLM 4.5 NCAR technical note, NCAR Tech Note, 2013.